# Why Linear Recurrent Memory Works in Partially Observable Reinforcement Learning

**Yike Zhao** [1]   **Onno Eberhard** [2]   **Malek Khammassi** [1]   **Ali H. Sayed** [1]   **Michael Muehlebach** [2]

## Abstract

The family of linear recurrent neural networks has shown strong performance as recurrent memory units in partially observable reinforcement learning. We provide a theoretical justification for their empirical effectiveness by constructing and studying two linear filters: (i) the first exactly reproduces the pre–softmax logits of the belief vector in a hidden Markov model (HMM) under a deterministic transition matrix, thereby serving as a sufficient statistic for optimal policy learning, (ii) the second achieves vanishing state-decoding error under a nearly deterministic transition matrix, thus reducing state ambiguity to near zero. The results extend to action-controlled HMMs, where the corresponding linear filters become time-varying with action-dependent dynamics. We illustrate our main results through numerical experiments and further show that the constructed linear filter serves as a strong feature extractor in a small reinforcement learning game.

## 1. Introduction

Many environments in reinforcement learning (RL) are partially observable. In this setting, formalized as a partially observable Markov decision process (POMDP), the true state of the environment is unknown and the agent only receives stochastic observations. Hence, a single observation does not fully determine the state and the agent must therefore leverage its interaction history to make effective decisions. A common approach is to learn a *recurrent memory* that recursively compresses this history into a latent state (Ni et al., 2022; Lu et al., 2023).

A particularly effective choice are *linear recurrent neural networks* (linear RNNs), where the hidden state transition

---
[1]EPFL, Lausanne, Switzerland [2]Max Planck Institute for Intelligent Systems, Tübingen, Germany. Correspondence to: Yike Zhao <yike.zhao@epfl.ch>.

*Proceedings of the 43$^{rd}$ International Conference on Machine Learning*, Seoul, South Korea. PMLR 306, 2026. Copyright 2026 by the author(s).

is linear and governed by a state transition matrix. Existing linear RNNs can be classified as either *time-invariant* (Gu et al., 2021; Smith et al., 2022; Orvieto et al., 2023) or *time-varying* (Gu & Dao, 2024; Yang et al., 2024a;b), depending on whether this matrix is fixed or input-dependent. Compared with nonlinear RNNs, linear RNNs achieve substantially lower training time by parallelizing computation across time steps (Gu et al., 2021; Gu & Dao, 2024), and have demonstrated strong performance on partially observable benchmarks (Lu et al., 2023; 2024). In contrast, nonlinear RNNs often suffer from optimization issues that lead to suboptimal results (Pascanu et al., 2013; Kanai et al., 2017).

A sufficient statistic for optimal decision-making in POMDP is the Bayesian posterior distribution over the environment's state given the history of interactions (e.g., Krishnamurthy, 2016). The posterior, or belief vector, is updated recursively following Bayes' rule, resulting in a nonlinear optimal filter. Therefore, a *linear* recurrent memory cannot in general reproduce these *nonlinear* dynamics. This raises questions about the information captured by linear dynamics and, more broadly, the explanation of the strong empirical success of linear RNNs. We address these questions by explicitly constructing linear filters related to the (nonlinear) log-belief dynamics in hidden Markov models (HMMs) and their action-controlled counterpart (the dynamics of general POMDPs). We establish the following two key results:

1. A time-invariant linear recurrent memory *exactly* reproduces the log-belief dynamics in HMMs with deterministic state transitions; the argument extends to POMDPs, yielding a time-varying linear memory.

2. We construct a linear recurrent memory called the time-invariant **Adaptive Logit Filter (ALF)** for HMMs with a nearly deterministic transition matrix, i.e., $T = D + \varepsilon Q$. This means that states evolve deterministically following $D$ and are perturbed by an arbitrary small stochastic element $\varepsilon Q$. We show that this filter recovers the true state with an asymptotic error rate of $\varepsilon \log(1/\varepsilon)$, which vanishes as $\varepsilon \to 0$. This rate matches the optimal asymptotic rate achieved by nonlinear filtering. A corresponding time-varying ALF satisfies the same guarantee in POMDPs.

Taken together, these results provide theoretical support that linear recurrent memories can represent or approximate the log-belief vector in partially observable RL. We underline these findings with experiments: We first illustrate the theoretical properties of ALF, using its time-invariant form, in a numerical simulation. We then construct a small RL environment and show that the time-varying ALF yields a strong learned policy and serves as an effective target for linear memories to learn.

The constructed filters offer new insights for the design of linear RNNs in partially observable RL:

1. Our two ALFs highlight a link between the eigenvalues of the latent dynamics matrix and environmental determinism: less stochastic environments require eigenvalues closer to the unit circle.

2. All constructed filters use a hidden dimension equal to the state space size, highlighting the representational efficiency achievable by linear memories.

3. The proposed two ALFs balance past memory and new information via the coefficients $(1 - \delta)$ and $\delta$, a structure widely observed in well-performing linear RNNs (e.g., Orvieto et al., 2023). Our analysis provides a principled interpretation of this mechanism in the reinforcement learning context.

## 2. Related Work

Our work focuses on providing a theoretical justification for linear RNNs in POMDPs. This section reviews common memory mechanisms in partially observable RL and prior works closely related to our theoretical developments.

The most widely used classes of memory are linear/nonlinear RNNs and Transformers. Although Transformers (Vaswani et al., 2017) revolutionized supervised sequence modeling, their impact in RL has been more modest (Lu et al., 2024) due to their nonrecurrent structure and the quadratic complexity of the attention mechanism. Existing theoretical studies in the context of RL have therefore focused on recurrent memory and investigated *specific* memory mechanisms, such as fixed-length windows (Efroni et al., 2022; Cayci et al., 2024) or exponential moving averages (Eberhard et al., 2025). We build on these works by providing a theoretical justification for the strong empirical success of linear RNNs in general, highlighting their validity as memory units.

We focus on the capability of linear RNNs for state tracking. Our analysis is inspired by the literature on Adaptive Social Learning (ASL; Bordignon et al., 2021; Khammassi et al., 2025), a multi-agent state tracking algorithm. In particular, Khammassi et al. (2025) analyze its performance under slow-varying HMMs. In the single-agent setting, we identify a direct connection between ASL and linear RNNs: after

removing normalization and applying a log transform, ASL falls into this class (Hu et al., 2023, (10) and (13)). Building on this insight, we construct linear filters that provably track the fast-changing states in nearly-deterministic HMMs and action-controlled HMMs.

## 3. Preliminaries

### 3.1. Notation

Boldface letters denote random variables. The floor and ceiling operators are denoted by $\lfloor \cdot \rfloor$ and $\lceil \cdot \rceil$, respectively. The operator $\text{diag}\{v\}$ transforms a state-related vector $v \in \mathbb{R}^N$ into a diagonal matrix, where the elements are interchangeably referred to as $v_i$ or $v_{x_i}$. Throughout the paper, $\{u_i\}$ denotes the canonical basis vectors, where the $i$-th component of $u_i$ equals 1 and all other components equal 0.

### 3.2. Linear Recurrent Memory

Linear recurrent memory can be broadly classified into two categories: *time-invariant* and *time-varying*. Time-invariant linear memory is widely used in RL practice, where the core dynamics take the form [1]:

$$z_k = A z_{k-1} + \phi(\boldsymbol{b}_k), \tag{1}$$

where $\boldsymbol{z}_k \in \mathbb{R}^H$ is the hidden state and $A \in \mathbb{R}^{H \times H}$ is the transition matrix. The input $\boldsymbol{b}_k$ is a feature vector from recent interactions, such as the current observation, an observation–action pair, or a fixed window of past observations. The encoder $\phi(\cdot)$ is a nonlinear mapping (e.g., a multi-layer perceptron (MLP)).

Time-varying linear memory has recently received increasing attention, whose dynamics become:

$$z_k = A(\boldsymbol{b}_k) z_{k-1} + \phi(\boldsymbol{b}_k), \tag{2}$$

where $A(\cdot)$ is a function that maps $\boldsymbol{b}_k$ to a matrix in $\mathbb{R}^{H \times H}$. This formulation retains parallelizable computation across time (Smith et al., 2022) while providing additional flexibility through input-dependent state transitions.

### 3.3. POMDPs

We consider a finite POMDP with state space $\mathcal{X} = \{x_1, \ldots, x_N\}$, action space $\mathcal{A} = \{a_1, \ldots, a_L\}$, and observation space $\mathcal{Y} = \{u_1, \ldots, u_S\}$, where each $u_i$ is a canonical basis vector in $\mathbb{R}^S$ (i.e., one-hot encoded). The initial state distribution is $\pi_0 \in \mathbb{R}^N$. Starting at time step $k = 0$, the true state $\boldsymbol{x}_k^\star$ evolves according to the controlled transition matrices $\{T(a)\}_{a \in \mathcal{A}}$, where

$$t_{ij}(a) \triangleq \mathbb{P}(\boldsymbol{x}_{k+1}^\star = x_i \mid \boldsymbol{x}_k^\star = x_j, \boldsymbol{a}_k = a). \tag{3}$$

---

[1] Linear time-invariant RNNs are often written as $\boldsymbol{z}_k = A\boldsymbol{z}_{k-1} + B\boldsymbol{b}_k$, but since inputs are typically passed through a nonlinear encoder, the effect of $B$ can be absorbed into $\phi$. The same argument also applies to the time-varying case.

Given the state–action pair $(\boldsymbol{x}_k^\star, \boldsymbol{a}_k)$, the agent receives a reward $r(\boldsymbol{x}_k^\star, \boldsymbol{a}_k)$. The next observation $\boldsymbol{y}_{k+1}$ is generated according to the emission matrix $E \in \mathbb{R}^{S \times N}$ with entries

$$e_{ij} \triangleq \mathbb{P}(\boldsymbol{y}_{k+1} = u_i \mid \boldsymbol{x}_{k+1}^\star = x_j). \tag{4}$$

Thus, the state–observation dynamics form an **action-controlled HMM** (Krishnamurthy, 2016).

### 3.4. Belief Vector and Logit–Space Filtering

The belief vector is the posterior distribution over the states given the entire action–observation history. It is a sufficient statistic for optimal decision-making (Krishnamurthy, 2016, Theorems 7.2.1, 7.6.1), and therefore provides a compact representation of the history. We denote the belief vector by $\boldsymbol{\alpha}_k \in \mathbb{R}^N$. In an action-controlled HMM, with $\alpha_0 = \pi_0$, for $k \geq 1$, $\boldsymbol{\alpha}_k$ has components:

$$\boldsymbol{\alpha}_k(x_i) \triangleq \mathbb{P}(\boldsymbol{x}_k^\star = x_i \mid \boldsymbol{y}_{1:k}, \boldsymbol{a}_{0:k-1}). \tag{5}$$

The belief vector can be updated recursively via Bayesian filtering, also known as the **optimal filter** (Shue et al., 1998, (3)):

$$\boldsymbol{\alpha}_k = \frac{\text{diag}\{E^\top \boldsymbol{y}_k\} T(\boldsymbol{a}_{k-1}) \boldsymbol{\alpha}_{k-1}}{\mathbb{1}^\top \text{diag}\{E^\top \boldsymbol{y}_k\} T(\boldsymbol{a}_{k-1}) \boldsymbol{\alpha}_{k-1}}. \tag{6}$$

Removing the normalization and taking elementwise logarithms on both sides yields:

$$\boldsymbol{w}_k^o = \log(T(\boldsymbol{a}_{k-1}) \exp(\boldsymbol{w}_{k-1}^o)) + \log(E^\top \boldsymbol{y}_k), \tag{7}$$

where $\boldsymbol{w}_k^o$ is related to the belief vector $\boldsymbol{\alpha}_k$ via softmax:

$$\boldsymbol{\alpha}_k = \text{softmax}(\boldsymbol{w}_k^o) \triangleq \frac{1}{\mathbb{1}^\top \exp(\boldsymbol{w}_k^o)} \exp(\boldsymbol{w}_k^o). \tag{8}$$

We therefore call $\boldsymbol{w}_k^o$ the **optimal logit** and (7) the **logit optimal filter (LOF)**.

From (7), we see that LOF looks similar to a linear recurrent memory, except for the nonlinear first term. In general,

$$\log(T(\boldsymbol{a}_{k-1}) \exp(\boldsymbol{w}_{k-1}^o)) \neq T(\boldsymbol{a}_{k-1}) \boldsymbol{w}_{k-1}^o, \tag{9}$$

so LOF is not a linear recurrent memory. Equality holds only in special cases, e.g., when $T(\boldsymbol{a}_{k-1})$ is the identity or a permutation matrix.

Motivated by this similarity, we focus on designing surrogate logit filters in the next two sections: recursions that evolve a surrogate logit $\boldsymbol{w}_k$ via a linear rule and provide the necessary information for learning a performant policy. The logit $\boldsymbol{w}_k$ induces a **proxy belief** via softmax,

$$\widehat{\boldsymbol{\alpha}}_k \triangleq \text{softmax}(\boldsymbol{w}_k). \tag{10}$$

*Remark* 3.1. For analytical convenience, we will mainly focus on the HMM setting with a fixed transition matrix $T$.

The concepts introduced above remain unchanged, except that (3), (5), (6), and (7) become action-independent. This structure underlies partially observable RL benchmarks (e.g., card games in POPGym (Morad et al., 2023)). Note that our key results extend naturally to the more general class of action-controlled HMMs. We will explicitly indicate whether the discussion pertains to the fixed-transition or action-controlled setting when necessary.

## 4. Deterministic Transitions

In this section, we construct linear recurrent memories that reproduce the optimal logits (i.e., $\boldsymbol{w}_k = \boldsymbol{w}_k^o$), which serve as sufficient statistics for optimal policy learning. We first consider a time-invariant linear filter in the HMM setting with deterministic transitions. We then extend the construction to the action-controlled HMM setting, resulting in a time-varying linear filter.

### 4.1. Linear Filters for HMMs

We consider a transition matrix $T$ of an HMM written in column form as:

$$T = \begin{bmatrix} T_1 & \cdots & T_N \end{bmatrix}, \tag{11}$$

where $T_i$ denotes the $i$-th column of $T$. The definition of a deterministic matrix is:

**Definition 4.1** (**Deterministic Matrix**). A matrix $T \in \mathbb{R}^{N \times N}$ is called **deterministic** if each column $T_i$ is one of the canonical basis vectors in $\mathbb{R}^N$:

$$T_i \in \{u_1, \ldots, u_N\}, \quad \forall i \in \{1, \ldots, N\}. \tag{12}$$

The definition of a deterministic matrix naturally includes permutation matrices, and it may also include matrices with repeated columns, which create transient states (i.e., states that, once exited, cannot be reentered).

**Lemma 4.2.** *Let $r$ denote the number of transient states for a deterministic transition matrix $T \in \mathbb{R}^{N \times N}$. The matrix $T$ can be represented as follows:*

$$T = \begin{bmatrix} T_{11} & 0 \\ T_{21} & P \end{bmatrix}, \tag{13}$$

*where:*

- *$T_{11} \in \mathbb{R}^{r \times r}$ is a strictly lower-triangular matrix with all-zero diagonal entries. The nilpotent index is denoted by $\ell$ (i.e., $T_{11}^\ell = 0$), where $\ell \leq r$. This matrix represents transitions between transient states.*
- *$T_{21} \in \mathbb{R}^{N-r \times r}$ represents transitions from transient states to recurrent states. If $r \neq 0$, it is non-zero.*
- *$P \in \mathbb{R}^{N-r \times N-r}$ describes the transitions among the recurrent states, and is a permutation matrix.*

*Proof.* See Appendix C.1. □

*Remark* 4.3. If $T$ is a permutation matrix, then the number of transient states is $r = 0$, and the nilpotent index is $\ell = 1$.

Now we reproduce the optimal logit $\boldsymbol{w}_k^o$ using a concrete realization of the time-invariant linear recurrent memory structure (1).

**Theorem 4.4.** *Under a deterministic transition matrix $T$ as defined in Definition 4.1, the optimal logits $\boldsymbol{w}_k^o$ can be reproduced by the following dynamics:*

$$\boldsymbol{w}_k = T'\boldsymbol{w}_{k-1} + \psi(k, \boldsymbol{y}_k, \ldots, \boldsymbol{y}_{k-\ell+1}), \quad (14)$$

*where*

$$T' \triangleq \begin{bmatrix} I_r & 0 \\ 0 & P \end{bmatrix}, \quad (15)$$

*$P$ is the same matrix as in (13), and $\psi(\cdot)$ is a nonlinear mapping that takes as input the most recent $\ell$ observations together with the time step $k$. The recursion is initialized at an arbitrary vector $w_0 \in \mathbb{R}^N$. The input to $\psi(\cdot)$ is padded with zeros for $k - \ell < 0$.*

*Proof.* Here we provide the proof when $T$ is a permutation matrix for intuition. The full proof for general deterministic matrices is deferred to Appendix C.2.

Let $T = P$, then we have $r = 0$ and $\ell = 1$. For $k \geq 1$, the optimal logit $\boldsymbol{w}_k^o$ satisfies:

$$\boldsymbol{w}_k^o \overset{(a)}{=} P\boldsymbol{w}_{k-1}^o + \log(E^\top \boldsymbol{y}_k), \quad (16)$$

where $(a)$ follows from the fact that $P \exp(\boldsymbol{w}_{k-1}^o)$ permutes the entries of $\exp(\boldsymbol{w}_{k-1}^o)$.

However, as the arbitrary initialization $w_0$ in our dynamics might differ from $w_0^o = \log(\pi_0)$, we design $\psi(k, y)$ to accommodate the potentially wrong initialization:

$$\psi(k, y) = \log\left(E^\top y\right) + \mathbb{I}\{k = 1\}P\left(\log(\pi_0) - w_0\right), \quad (17)$$

where $\mathbb{I}\{\cdot\}$ is the indicator function. Here, the effect of wrong initialization is canceled out when $k = 1$, and later on, only the first term remains valid. $\square$

Note that the structure of $\psi$ is, however, uncommon in deep learning. Understanding how well standard architectures, such as MLPs, can approximate this mapping is an interesting future direction, particularly given the common use of MLP encoders.

### 4.2. Linear Filters for Action-Controlled HMMs

We consider an action-controlled HMM and suppose that the transition matrices are permutation matrices, i.e., $T(a) = P(a), \forall a \in \mathcal{A}$. We then construct a realization of the time-varying linear recurrent structure (2) that reproduces the optimal logits.

**Corollary 4.5.** *If the transition matrices satisfy $T(a) = P(a), \forall a \in \mathcal{A}$, the optimal logits $\boldsymbol{w}_k^o$ can be reproduced by the following dynamics:*

$$\boldsymbol{w}_k = P(\boldsymbol{a}_{k-1})\boldsymbol{w}_{k-1} + \psi(k, \boldsymbol{y}_k, \boldsymbol{a}_{k-1}), \quad (18)$$

*where $\psi(\cdot)$ is a nonlinear mapping that takes as input the most recent action-observation pair together with the time step $k$. The initialization $w_0 \in \mathbb{R}^N$ is arbitrary.*

*Proof.* The result follows directly from Theorem 4.4 by replacing the fixed permutation matrix $P$ in (16)–(17) by $P(a)$ and $\psi(k, y)$ by $\psi(k, y, a)$. $\square$

Here, only the latest observation and action are required, since there are no transient states (i.e., $r = 0$) for each $P(a)$. It should be noted that both the constructed linear filters (14) and (18) have a hidden dimension $N$, the state space dimension. The choice of initialization $w_0$ is not critical, as is typical in linear RNNs, which often use zero initialization across tasks (Gu et al., 2021). Additionally, all eigenvalues of $T'$ and $P(\boldsymbol{a}_{k-1})$ lie on the unit circle, consistent with the requirements of linear RNNs for modeling long-range dependencies (Orvieto et al., 2023; Grazzi et al., 2024).

## 5. Nearly-Deterministic Transitions

In this section, we show that vanishing state-decoding error can be achieved by concrete realizations of linear recurrent memory. We first present a time-invariant filter for HMMs with nearly deterministic transitions, and then extend the construction to the action-controlled setting, yielding a time-varying linear filter.

The ability to reproduce the true states with vanishing error is important for effective policy learning. The benefits are further illustrated through experiments in Section 6.2.

We define the instantaneous state-decoding error at time step $k$ as:

$$p_k \triangleq \mathbb{P}\left[\arg\max_{x \in \mathcal{X}} \widehat{\boldsymbol{\alpha}}_k(x) \neq \boldsymbol{x}_k^\star\right]$$

$$\overset{(a)}{=} \mathbb{P}\left[\arg\max_{x \in \mathcal{X}} \boldsymbol{w}_k(x) \neq \boldsymbol{x}_k^\star\right]. \quad (19)$$

where $(a)$ follows from the fact that softmax preserves the ordering of elements. As a reference, the maximum a posteriori rule based on the exact belief vector $\boldsymbol{\alpha}_k$ in (6) achieves the minimum state-decoding error. We refer to this rule as the *Bayes-optimal decoder*, and compare the performance of our filters against it.

### 5.1. Linear Filters for HMMs

**Time-Invariant ALF.** In the following, we construct a time-invariant linear filter under the nearly-deterministic

transition model. A nearly-deterministic transition matrix in an HMM is defined as follows:

**Definition 5.1** (**Nearly-Deterministic Matrix**). Let $D$ be a deterministic transition matrix as in Definition 4.1. A matrix $T$ is called **nearly-deterministic** if it admits the decomposition

$$T = D + \varepsilon Q \qquad (20)$$

for some *small* drift parameter $0 < \varepsilon < 1$.

The matrix $Q$ satisfies for each $j = 1, \ldots, N$:

$$q_{ij} \geq 0 \;\; (i \neq n_j), \quad q_{n_j j} = -\sum_{i \neq n_j} q_{ij}, \qquad (21)$$

where $n_j$ is the row index of the unique 1 in column $j$ of $D$ (i.e., $d_{n_j j} = 1$, $d_{ij} = 0$, for all $i \neq n_j$).

The condition (21) guarantees that the matrix $T$ is a valid transition matrix (necessary and sufficient). It can be easily shown that $T$ is column-stochastic with nonnegative entries. This type of matrix is called *nearly-deterministic* because the small parameter $\varepsilon$ allows the deterministic component $D$ to dominate. As a result, the states $\boldsymbol{x}_k^\star$ follow the deterministic dynamics induced by $D$ for most time steps, with only occasional irregular transitions due to the term $\varepsilon Q$.

*Remark* 5.2. When $D = I$, $T$ in Definition 5.1 reduces to the transition matrix of the slowly varying Markov chains studied by Khasminskii & Zeitouni (1996), Yin & Krishnamurthy (2005), and Khammassi et al. (2025).

According to Lemma 4.2, the deterministic transition matrix $D$ can be expressed, after proper state labeling, as

$$D = \begin{bmatrix} P & D_{12} \\ 0 & D_{22} \end{bmatrix}, \qquad (22)$$

where $P \in \mathbb{R}^{N-r \times N-r}$ is a permutation matrix, $D_{12} \in \mathbb{R}^{N-r \times r}$, and $D_{22} \in \mathbb{R}^{r \times r}$ is nilpotent with index $\ell$. In contrast to Lemma 4.2, we label the recurrent states of $D$ as the first $N - r$ states:

$$\widetilde{\mathcal{X}} \triangleq \{x_1, \ldots, x_{N-r}\}, \qquad (23)$$

and the transient states as the last $r$ states, i.e., $\mathcal{X} \setminus \widetilde{\mathcal{X}}$. In what follows, we assume $D$ has this form.

Motivated by the ASL algorithm in Bordignon et al. (2021); Khammassi et al. (2025), we propose the following realization of the time-invariant linear recurrent memory (1) in the logit space, referred to as the **adaptive logit filter (ALF)** for nearly-deterministic transition matrices:

$$\boldsymbol{w}_k = \begin{bmatrix} P & 0 \\ 0 & I_r \end{bmatrix} (1 - \delta)\boldsymbol{w}_{k-1} + \delta \log(E^\top \boldsymbol{y}_k). \quad (24)$$

ALF recursively updates a surrogate logit vector by combining a permuted, discounted past surrogate logit with the instantaneous log-likelihood of new observations. When $\delta = 0$, ALF depends entirely on the past surrogate logit and thus cannot adapt to changes in the underlying observation distribution that arise from irregular state transitions. In contrast, when $\delta = 1$, ALF relies solely on the instantaneous observation, allowing ALF to adapt quickly but making it more sensitive to observation noise. Hence, $\delta$ governs the speed of adaptation, trading off stability against responsiveness.

The main difference in relation to the model used by Bordignon et al. (2021) and Khammassi et al. (2025) is the presence of the matrix before $(1 - \delta)\boldsymbol{w}_{k-1}$ in (24). This matrix enables $\boldsymbol{w}_k$ to capture the fast state transition that dominates most of the time; without it, the state decoding error increases substantially. For $\boldsymbol{w}_k$, we denote the components corresponding to the recurrent states as:

$$\widetilde{\boldsymbol{w}}_k \triangleq \begin{bmatrix} \boldsymbol{w}_k(x_1) \\ \vdots \\ \boldsymbol{w}_k(x_{N-r}) \end{bmatrix} \in \mathbb{R}^{N-r}. \qquad (25)$$

We impose the following assumption on the initial condition of ALF.

**Assumption 5.3.** The initial $w_0$ for ALF is:

$$w_0 = \begin{bmatrix} \widetilde{w}_0 \\ -\infty \mathbb{1}_r \end{bmatrix}, \qquad (26)$$

where $\widetilde{w}_0$ is finite (i.e., $\|\widetilde{w}_0\|_\infty < \infty$).

We adopt the convention $-\infty \cdot 0 = 0$. Assumption 5.3 ensures that the decoded states over time are restricted to the recurrent states, never transient ones.

**Observability Assumption.** Our goal is to analyze the state-decoding property of ALF, which will require additional definitions and assumptions on the emission matrix. These will be introduced next.

For the permutation $P$, we define its order as $M$:

**Definition 5.4.** The order of a permutation matrix $P$ is the smallest positive integer $M$ such that $P^M = I$.

**Lemma 5.5.** *(Humphreys & Prest, 2004, Theorem 4.2.6) Every permutation matrix has a finite order $M$, which is given by the least common multiple of the lengths of the cycles in the underlying permutation.*

For any $x \in \widetilde{\mathcal{X}}$, let $\sigma : \widetilde{\mathcal{X}} \to \widetilde{\mathcal{X}}$ denote the permutation operation corresponding to $P$. Applying the permutation matrix $P$ repeatedly $i$ times corresponds to $\sigma^i(x)$. Note that $\sigma^M(x) = x$ according to Lemma 5.5.

The emission matrix $E \in \mathbb{R}^{S \times N}$ has the following form:

$$E = \begin{bmatrix} E_{1:N-r} & E_{N-r+1} & \dots & E_N \end{bmatrix} \quad (27)$$

where $E_1, \dots, E_N$ are column vectors. We denote

$$\widetilde{E} \triangleq E_{1:N-r} \quad (28)$$

and define a corresponding stacked matrix $\widetilde{E}_s$ as:

$$\widetilde{E}_s \triangleq \begin{bmatrix} \widetilde{E}P^0 \\ \vdots \\ \widetilde{E}P^{M-1} \end{bmatrix} \in \mathbb{R}^{SM \times N-r}, \quad (29)$$

where $M$ is the order of $P$ defined in Definition 5.4. To build intuition, we consider the first column:

$$\widetilde{E}_s u_1 = \begin{bmatrix} \widetilde{E}P^0 u_1 \\ \vdots \\ \widetilde{E}P^{M-1} u_1 \end{bmatrix} \overset{(a)}{=} \begin{bmatrix} \widetilde{E}u_{x_1} \\ \vdots \\ \widetilde{E}u_{\sigma^{M-1}(x_1)} \end{bmatrix}, \quad (30)$$

where $(a)$ follows since $P^i u_1$ applies the permutation $\sigma$ to $x_1$ exactly $i$ times, yielding $u_{\sigma^i(x_1)}$. The first $S$ elements correspond to the observation distribution when the true state is $x_1$, the next $S$ elements correspond to the state $\sigma(x_1)$, and so on.

**Assumption 5.6.** The emission matrix $E$ satisfies:

(a) All elements of $\widetilde{E}$ are strictly positive.

(b) No two columns of the stacked matrix $\widetilde{E}_s$ are identical.

In Assumption 5.6, no restrictions are placed on the observation models $\mathbb{P}(\cdot \mid x)$ for transient states $x \in \mathcal{X} \setminus \widetilde{\mathcal{X}}$. For recurrent states $x \in \widetilde{\mathcal{X}}$, Assumption 5.6a requires that the distributions $\mathbb{P}(\cdot \mid x)$ share identical support, ensuring finite Kullback–Leibler divergence across states, while Assumption 5.6b enforces that observation distributions along different $M$-step trajectories are distinct. Similar multi-step observability assumptions are commonly imposed in POMDPs (e.g., Liu et al., 2022; Golowich et al., 2023) to ensure stable filtering and learnability. Assumption 5.6 is weaker than the one used in Khammassi et al. (2025); we defer the discussion to Appendix A.1.

**Main Results.** Our goal is to recover the true state $x_k^\star$. The following theorem provides a sufficient condition on the parameters $\varepsilon$ and $\delta$ under which the dynamics in (24) achieve vanishing error probability in identifying $x_k^\star$. This result extends Theorems 1 and 2 of Khammassi et al. (2025) to the model (24), allowing a more general class of transition matrices and relying on a weaker observability assumption. Further intuition on the sufficient condition in the special case of $D = I$ can be found in Khammassi et al. (2025).

**Theorem 5.7.** *Consider a nearly-deterministic transition matrix $T$ under Assumptions 5.3 and 5.6. Let $\delta = \delta_\varepsilon$ be the adaptation parameter used for a given drift parameter $\varepsilon$. If the following conditions are satisfied:*

$$\lim_{\varepsilon \to 0} \delta_\varepsilon = 0, \quad (31)$$

$$\lim_{\varepsilon \to 0} \frac{\varepsilon}{\delta_\varepsilon} = 0, \quad (32)$$

*then the **ALF** consistently learns the true states as $\varepsilon$ goes to zero, which is formally expressed as*

$$\lim_{\varepsilon \to 0} \limsup_{k \to \infty} p_k = 0. \quad (33)$$

*Moreover, following conditions (31) and (32), consider*

$$\delta_\varepsilon = \frac{\lambda}{\log(1/\varepsilon)}, \quad \text{with } \lambda \in (0, \xi), \quad (34)$$

*where the constant $\xi > 0$ depends only on the permutation backbone $P$ and the likelihood model $E$; its detailed definition and further discussion are provided in Appendices D.1 and D.2, respectively. Then it holds that*

$$\limsup_{k \to \infty} p_k \leq \kappa \, \varepsilon \log \frac{1}{\varepsilon} + o(\varepsilon \log \frac{1}{\varepsilon}), \quad (35)$$

*with the constant $\kappa > 0$ defined in Appendix D.1. The resulting decay rate matches that of the Bayes-optimal decoder (Khasminskii & Zeitouni, 1996), in the sense that the long-term error is also dominated by an $\varepsilon \log(1/\varepsilon)$ term.*

*Proof.* Let $\varepsilon > 0$ and $k \in \mathbb{N}$. For any $T_\varepsilon > 0$, we can write the instantaneous error probability as follows:

$$p_k = \mathbb{P}\left[ \arg\max_{x \in \mathcal{X}} \boldsymbol{w}_k(x) \neq \boldsymbol{x}_k^\star, \text{ no IrJ in } [k - T_\varepsilon, k] \right]$$
$$+ \mathbb{P}\left[ \arg\max_{x \in \mathcal{X}} \boldsymbol{w}_k(x) \neq \boldsymbol{x}_k^\star, \text{ IrJ in } [k - T_\varepsilon, k] \right]$$
$$\leq \mathbb{P}\left[ \arg\max_{x \in \mathcal{X}} \boldsymbol{w}_k(x) \neq \boldsymbol{x}_k^\star, \text{ no IrJ in } [k - T_\varepsilon, k] \right]$$
$$+ \mathbb{P}\left[ \text{IrJ in } [k - T_\varepsilon, k] \right], \quad (36)$$

where "IrJ" stands for *irregular jumps*. An irregular jump is observed at instant $k$ if $\boldsymbol{x}_{k-1}^\star$ to $\boldsymbol{x}_k^\star$ does not follow matrix $D$. We would like to properly choose $T_\varepsilon$ such that both terms in (36) are small. For each $\delta_\varepsilon$, choose

$$T_\varepsilon \triangleq \frac{\alpha}{\delta_\varepsilon}, \quad (37)$$

where $\alpha > 0$. As $\varepsilon$ goes to 0, $\delta_\varepsilon$ goes to 0 as well according to (31). For $\delta_\varepsilon$ small enough, it is possible to find a $h_\varepsilon > 0$ such that

$$h_\varepsilon \triangleq \left\lfloor \frac{T_\varepsilon}{M} \right\rfloor, \quad (38)$$

which means $T_\varepsilon = \frac{\alpha}{\delta_\varepsilon} \geq h_\varepsilon M$, where $M$ is the order of the permutation matrix $\tilde{P}$ according to Lemma 5.5. We denote

$$I_k \triangleq [k - h_\varepsilon M + 1, \, k] \qquad (39)$$

Since $\{\text{no IrJ in } [k - T_\varepsilon, k]\} \subseteq \{\text{no IrJ in } I_k\}$, we further upper bound (36) by:

$$\sum_{x' \in \mathcal{X}} \mathbb{P}\left[\arg\max_{x \in \mathcal{X}} \boldsymbol{w}_k(x) \neq \boldsymbol{x}_k^\star \middle| \text{no IrJ in } I_k, \boldsymbol{x}_{k-h_\varepsilon M}^\star = x'\right]$$
$$\mathbb{P}\left[\text{no IrJ in } I_k \middle| \boldsymbol{x}_{k-h_\varepsilon M}^\star = x'\right] \mathbb{P}\left[\boldsymbol{x}_{k-h_\varepsilon M}^\star = x'\right]$$
$$+ \mathbb{P}\left[\text{IrJ in } [k - T_\varepsilon, k]\right]$$
$$\overset{(a)}{\leq} \sum_{x' \in \widetilde{\mathcal{X}}} \mathbb{P}\left[\arg\max_{x \in \widetilde{\mathcal{X}}} \widetilde{\boldsymbol{w}}_k(x) \neq \boldsymbol{x}_k^\star \middle| \text{no IrJ in } I_k, \boldsymbol{x}_{k-h_\varepsilon M}^\star = x'\right]$$
$$+ \sum_{x' \in \mathcal{X} \setminus \widetilde{\mathcal{X}}} \mathbb{P}\left[\boldsymbol{x}_{k-h_\varepsilon M}^\star = x'\right] + \mathbb{P}\left[\text{IrJ in } [k - T_\varepsilon, k]\right], \qquad (40)$$

where $(a)$ follows since probabilities are smaller or equal to 1 and $\arg\max_{x \in \mathcal{X}} \boldsymbol{w}_k(x) = \arg\max_{x \in \widetilde{\mathcal{X}}} \widetilde{\boldsymbol{w}}_k(x)$ due to the $-\infty$ initialization of $w_0$ for the transient states, as specified in Assumption 5.3.

From Lemma D.1, since $h_\varepsilon M$ does not change as $k \to \infty$, we have:

$$\lim_{\varepsilon \to 0} \limsup_{k \to \infty} \sum_{x' \in \mathcal{X} \setminus \widetilde{\mathcal{X}}} \mathbb{P}\left[\boldsymbol{x}_{k-h_\varepsilon M}^\star = x'\right] = 0, \qquad (41)$$

$$\lim_{\varepsilon \to 0} \limsup_{k \to \infty} \mathbb{P}\left[\text{IrJ in } [k - T_\varepsilon, k]\right] = 0. \qquad (42)$$

By Lemma D.2, under the specified evolution pattern of the true states, we have:

$$\lim_{\varepsilon \to 0} \limsup_{k \to \infty} \mathbb{P}\left[\arg\max_{x \in \widetilde{\mathcal{X}}} \widetilde{\boldsymbol{w}}_k(x) \neq \boldsymbol{x}_k^\star \middle| \text{no IrJ in } I_k,\right.$$
$$\left. \boldsymbol{x}_{k-h_\varepsilon M}^\star = x'\right] = 0, \, \forall x' \in \widetilde{\mathcal{X}} \qquad (43)$$

Now we return to (40) and complete the proof:

$$\lim_{\varepsilon \to 0} \limsup_{k \to \infty} p_k \overset{(a)}{=} 0 \qquad (44)$$

where $(a)$ follows from (41), (42), and (43). The derivations of Lemmas D.1 and D.2, as well as the error decaying rate under the $\delta_\varepsilon$ rule (34) can be found in Appendix D.1. □

Theorem 5.7 shows that, by appropriately controlling the adaptation parameter, linear recurrent memory can achieve vanishing state-decoding error, with a decay rate that matches that of the Bayes-optimal decoder. This result provides a quantitative and strong performance bound on ALF's state-decoding capability, demonstrating that the gap between ALF and the nonlinear optimal filter is small.

## 5.2. Linear Filters for Action-Controlled HMMs

In action-controlled HMMs, we assume each transition matrix is nearly a permutation:

$$T(a) = P(a) + \varepsilon Q(a), \quad a \in \mathcal{A}, \qquad (45)$$

with each $T(a)$ being a subclass of nearly-deterministic matrices. We propose the following time-varying ALF:

$$\boldsymbol{w}_k = P(\boldsymbol{a}_{k-1})(1 - \delta)\,\boldsymbol{w}_{k-1} + \delta \log(E^\top \boldsymbol{y}_k), \qquad (46)$$

Moreover, we assume the following:

**Assumption 5.8.** The initial and emission matrix $E$ satisfy:

1. The initialization $w_0$ is finite, i.e., $\|w_0\| < \infty$.

2. All entries of $E$ are strictly positive, and no two columns of $E$ are identical.

**Theorem 5.9.** *Consider transition matrices satisfying* (45). *Under Assumption 5.8, the results in Theorem 5.7 hold for time-varying ALF.*

*Proof.* The result follows analogously to the proof of Theorem 5.7. □

Our results show the ability of time-varying ALF in recovering the true states. Experiments in the next section underline that time-varying ALF enables learning performant policies. Moreover, both the time-invariant and time-varying ALF resemble practical linear RNNs, with eigenvalues near the unit circle that support long-range dependencies.

## 6. Experiments

In this section, we first illustrate the theoretical properties of ALF with a numerical example in the HMM setting. We then introduce a partially observable game, the *RingWorld*, to show that time-varying ALF is a suitable learning target for linear RNNs.

*Remark* 6.1. In this section, we use "ALF" for both its time-invariant and time-varying variants for brevity.

### 6.1. Illustrative Example

We consider a HMM with state space $\mathcal{X} = \{x_1, x_2\}$ and observation space $\mathcal{Y} = \{u_1, u_2\}$. Following Definition 5.1, the nearly-deterministic transition matrix $T$ and the emission matrix $E$ are:

$$T = \begin{bmatrix} \varepsilon & 1 - \varepsilon \\ 1 - \varepsilon & \varepsilon \end{bmatrix}, \quad E = \begin{bmatrix} 0.9 & 0.1 \\ 0.1 & 0.9 \end{bmatrix}. \qquad (47)$$

Since $D$ is a two-state permutation matrix, we have $r = 0$, and ALF (24) simplifies accordingly. Similar to Khammassi et al. (2025), we select the following $\delta_\varepsilon$ values that satisfy conditions (31) and (32).

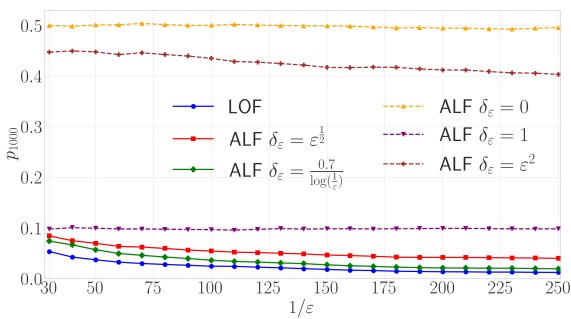

*Figure 1.* Long-run error probability versus $1/\varepsilon$ for ALF and LOF. Solid/dashed lines correspond to valid/invalid $\delta_\varepsilon$ selections, respectively.

$$\delta_\varepsilon = \varepsilon^{1/2}, \quad \delta_\varepsilon = \frac{0.7}{\log(1/\varepsilon)}. \tag{48}$$

The latter choice also satisfies (34); the detailed derivations are provided in Appendix D.2. In addition to valid $\delta_\varepsilon$ choices, we also consider three invalid ones:

$$\delta_\varepsilon = \varepsilon^2, \quad \delta_\varepsilon = 0, \quad \delta_\varepsilon = 1, \tag{49}$$

where $\delta_\varepsilon = \varepsilon^2$ and $\delta_\varepsilon = 0$ violate (32), and $\delta_\varepsilon = 1$ violates both (31) and (32). We choose 23 values of $1/\varepsilon$ uniformly from $[30, 250]$ and run 20,000 Monte Carlo iterations for each $\varepsilon$ to estimate the instantaneous error probabilities. As the error stabilizes after 1000 steps (see Appendix E.1), we adopt $p_{1000}$ as an empirical proxy for $\limsup_{k\to\infty} p_k$ and study its variation as $1/\varepsilon$ increases, as shown in Figure 1. Valid choices yield vanishing long-run error probabilities as $\varepsilon \to 0$, with only a small performance gap. Consistently, $\delta_\varepsilon = 0.7/\log(1/\varepsilon)$ performs better than $\delta_\varepsilon = \varepsilon^{1/2}$. For invalid choices, $p_{1000}$ does not vanish; for example, $\delta_\varepsilon = \varepsilon^2$ is too small to track the state transitions. Together, these observations provide evidence that Theorem 5.7 accurately describes the state-decoding in practice. These observations are also consistent with those reported for $D = I$ in Khammassi et al. (2025).

### 6.2. RingWorld Game

In this experiment, we study a general POMDP with action-dependent state transitions. Motivated by a classical aliased cyclic-state benchmark from the predictive modeling literature (e.g., Tanner & Sutton, 2005; Schlegel et al., 2023), we design a *RingWorld* game with stochastic observations and more complex action-dependent dynamics.

The state space is $\mathcal{X} = \{1, \ldots, 12\}$ (positions on a ring) and the action set is $\mathcal{A} = \{\text{CW1}, \text{CW2}, \text{CCW1}, \text{CCW2}\}$, where CW1/CCW1 move one step clockwise/counterclockwise. The agent may overshoot or slip, each with probability $\varepsilon = 0.05$, making the transitions nearly permutation. Four beacons are placed around the ring. At each step, the agent observes the index of the strongest beacon, encoded as $u_i \in \mathcal{Y} = \{u_1, \ldots, u_4\}$. The closer the agent is to a beacon,

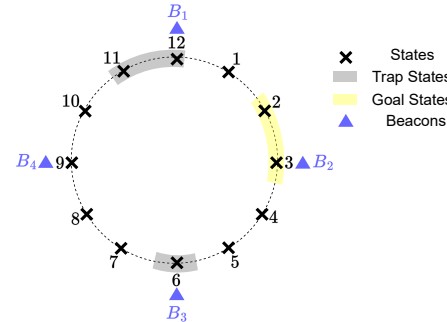

*Figure 2.* RingWorld environment. $B_i$ represents the $i$-th beacon.

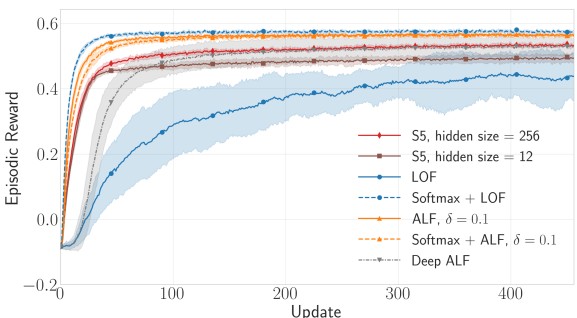

*Figure 3.* Episodic return versus update. The shaded area is the standard deviation across 8 random seeds. ALF/LOF are constructed from environment knowledge, whereas S5 and Deep ALF are trained from scratch.

the more likely it will be observed. The states $6, 11$, and $12$ are "trap" states, and the states $2$ and $3$ are goal states. Entering a trap state incurs a penalty, while entering a goal state yields a reward. An episode terminates after a fixed number of steps and the total episodic return lies in $[-1, 1]$. The objective is to avoid or quickly escape the trap states while remaining in the goal states under partial observability. The setting is illustrated in Figure 2.

We train Proximal Policy Optimization agents (PPO; Schulman et al., 2017) to act in this environment under four types of memory: **LOF**, **ALF**, an **S5-based memory** which is a single S5 layer (Smith et al., 2022) with a two–layer MLP encoder $\phi(\cdot)$, and an ALF-inspired deep learning variant called **Deep ALF**. Specifically, **Deep ALF** is obtained by parameterizing the ALF recursion in complex space and treating the parameters as learnable, similar to standard approaches for designing linear RNNs. For **LOF** and **ALF**, the memory output is either fed directly to the actor and critic, or passed through a softmax to form a (proxy) belief vector. Both the actor and critic are implemented as two–layer MLPs. For **LOF** and **ALF**, the memory is fixed and obtained from environment knowledge (**ALF** follows (46), while **LOF** uses (7)), and only the actor and critic are trained. For both **S5-based memory** and **Deep ALF**, the memory parameters are randomly initialized and jointly trained with the actor and critic.

As shown in Figure 3, **ALF** with direct outputs converges quickly to a strong return, whereas **LOF** trains slowly with high variance across seeds. The instability stems from the logits in **LOF** drifting increasingly negative over time, while **ALF** remains bounded (Lemma D.11). Applying a softmax to **LOF** recovers a belief vector, stabilizes training, and yields the best returns in this experiment. For the **S5-based memory**, increasing the hidden size from 12 (the memory size of **LOF**, **ALF**, and **Deep ALF**) to 256 allows it to match the performance of **Deep ALF**, but it still falls short of **ALF** and softmax-processed **LOF**. Notably, **Deep ALF** is more parameter-efficient than the **S5-based memory**, as its encoder has a number of parameters comparable to matrix $E$, rather than a two-layer MLP. Overall, these results show that **ALF** provides an efficient and reliable linear memory, and **Deep ALF** achieves better performance than the **S5-based memory** using fewer parameters.

Moreover, when environment knowledge is partially or fully available, **Deep ALF** can leverage this information to obtain improved initialization and achieve higher returns (see Figure 10). In contrast, it remains unclear how such prior knowledge can be incorporated into existing linear recurrent networks. More details can be found in Appendix E.2.

## 7. Discussion and Limitations

Our analysis focuses on environments with (near) deterministic transitions, which provide a sufficient regime for linear memory to perform well. While this regime may appear restrictive, such environments are not uncommon in existing benchmarks (e.g., the Passive T-Maze task in Lu et al. (2024) and the RepeatPrevious task in Lu et al. (2023)). Moreover, although our theoretical results are developed for finite state, observation, and action spaces, many continuous-control problems can be approximated by finite, nearly deterministic POMDPs through discretization (e.g., robotic control tasks in MuJoCo (Todorov et al., 2012)). Extending the evaluation to these more complex environments is an important direction for future work.

A natural question is how linear memory performs outside the (near) deterministic regime. We note that, in *highly stochastic* environments, sequence models may offer limited gains over memoryless models for state decoding. For example, in the fully stochastic version of the two-state example from Section 6.1 ($\varepsilon = 0.5$; see (47)), the first term in the LOF update (7) becomes proportional to the all-one vector, and the optimal filter reduces to a myopic estimator based solely on $\log(E^\top \boldsymbol{y})$. Consequently, past observations no longer provide useful information for state estimation. This suggests that, when the environment dynamics are sufficiently stochastic, leveraging long-term history may provide little benefit, regardless of whether one uses linear RNNs or other sequence models.

## 8. Conclusions

We justify the use of linear recurrent neural networks (RNNs) in partially observable reinforcement learning by constructing specific instances: First, we show that linear recurrent memory can exactly reproduce the log-belief dynamics under deterministic state transitions. Second, we introduce the Adaptive Logit Filter (ALF), which asymptotically recovers the true states under nearly deterministic state transitions. Theoretical properties of **ALF** are illustrated through a numerical example, and we further demonstrate that **ALF** provides a strong learning target in a small POMDP environment, the *RingWorld*.

## Impact Statement

This paper presents work whose goal is to advance the field of machine learning. There are many potential societal consequences of our work, none which we feel must be specifically highlighted here.

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

# A. Additional Discussion

## A.1. Observability Assumption

When $r = 0$ and $P = I$, we have $\widetilde{E} = E$ and $\widetilde{E}_s$ reduces to $E$ since $M = 1$. In this case, Assumptions 5.6a and 5.6b coincide with those of Khammassi et al. (2025, Assumptions 1 and 3). For $r \geq 1$ or $M > 1$, however, they are weaker: if $r \geq 1$, the columns $E_{N-r+1}, \ldots, E_N$ may be duplicates of others or contain zero components; if $M > 1$, $\widetilde{E}$ may contain identical columns. For example:

*Example* 1. Let

$$P = \begin{bmatrix} 0 & 1 & 0 & 0 \\ 1 & 0 & 0 & 0 \\ 0 & 0 & 0 & 1 \\ 0 & 0 & 1 & 0 \end{bmatrix}, \quad \widetilde{E} = \begin{bmatrix} 0.5 & 0.8 & 0.5 & 0.2 \\ 0.5 & 0.2 & 0.5 & 0.8 \end{bmatrix}. \tag{50}$$

Then,

$$\widetilde{E}_s = \begin{bmatrix} 0.5 & 0.8 & 0.5 & 0.2 \\ 0.5 & 0.2 & 0.5 & 0.8 \\ 0.8 & 0.5 & 0.2 & 0.5 \\ 0.2 & 0.5 & 0.8 & 0.5 \end{bmatrix}, \tag{51}$$

where $\widetilde{E}$ assigns identical observation distributions to $x_1$ and $x_3$, yet no two columns of $\widetilde{E}_s$ are identical.

## A.2. Beyond Permutation-Based POMDPs

For general nearly-deterministic transition matrices (instead of nearly-permutation), there's no easy extension of ALF. Our approach relies on setting the logit entries corresponding to transient states to $-\infty$. Consider a simple deterministic experiment:

$$T(a_1) = \begin{bmatrix} 1 & 1 \\ 0 & 0 \end{bmatrix}, \quad T(a_2) = \begin{bmatrix} 0 & 0 \\ 1 & 1 \end{bmatrix}, \tag{52}$$

In this example, action $a_1$ makes $x_2$ transient, while action $a_2$ makes $x_1$ transient. If one action is applied consistently, the model reduces to an HMM, where setting the corresponding transient state to $-\infty$ is valid. However, if actions alternate, no true transient states exist (since every state can be visited with non-negligible probability), and there is no natural way to extend ALF to this case.

Even when transient states coincide across transition matrices, generalization is still difficult. Consider Figure 4: the transient states are always $x_1$ and $x_2$, but under alternating actions $a_1, a_2, a_1, \ldots$, a nonzero initial probability for $x_1$ implies that the probability for $x_1$ and $x_2$ never vanishes.

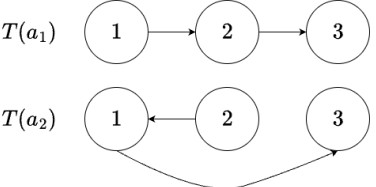

*Figure 4.* Illustration of a difficult case for generalization.

A possible generalization to nearly-deterministic POMDPs is to assume backbone deterministic matrices

$$D(a) = \left[ \begin{array}{c|c} P(a) & D_{12}(a) \\ \hline 0 & D_{22}(a) \end{array} \right], \tag{53}$$

with strictly upper-triangular $D_{22}(a)$ for all $a \in \mathcal{A}$, so that transient-state probabilities go to zero after finitely many steps.

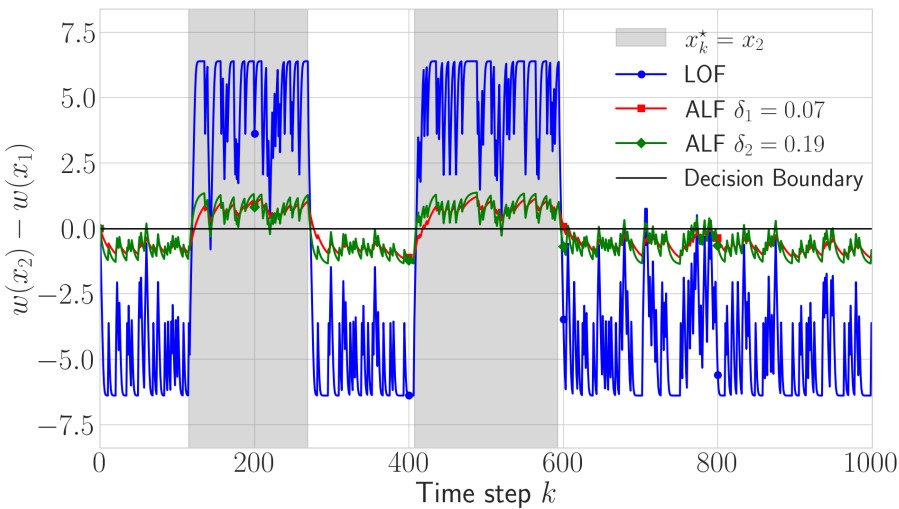

*Figure 5.* Sample evolution of the logit difference $\boldsymbol{w}_k(x_2) - \boldsymbol{w}_k(x_1)$ for **LOF** and **ALF** under $\varepsilon = 0.005$.

## B. Visualization of ALF's State Tracking

In this section, we provide additional visualizations to further illustrate how ALF recovers the underlying states.

### B.1. Time-Invariant ALF

In this section, we consider a two-state HMM:

$$T = \begin{bmatrix} 1 - \varepsilon & \varepsilon \\ \varepsilon & 1 - \varepsilon \end{bmatrix}, \quad E = \begin{bmatrix} 0.8 & 0.2 \\ 0.2 & 0.8 \end{bmatrix}, \tag{54}$$

with $\varepsilon = 0.005$. In this setting, the sign of the difference $\boldsymbol{w}_k(x_2) - \boldsymbol{w}_k(x_1)$ fully determines the decoded state (positive implies $x_2$, and vice versa). Consider $\delta_1 = 0.07$ and $\delta_2 = 0.19$.

Two observations can be drawn from Figure 5. First, although the overall trends are similar, there is a clear discrepancy between the dynamics of **ALF** and **LOF**. Nevertheless, **ALF** still achieves good state decoding. Second, since $\delta_1 < \delta_2$, the dynamics under $\delta_1$ appear smoother than those under $\delta_2$, as the smaller value reduces sensitivity to instantaneous (and noisy) observations. However, this smoother behavior comes at the cost of slower adaptation to irregular state transitions, as reflected by the lag after switching points (e.g., $k = 115, 270, 406, 593$). This reflects the intrinsic tradeoff between stability and responsiveness in **ALF**. Similar plots can be generated for the model in Section 6.1; however, due to the frequent state transitions, these are less interpretable.

### B.2. Time-Varying ALF

We next consider the RingWorld environment in Section 6.2. Figure 6 shows a sample trajectory of true states, estimated states, actions, and observations. **ALF** is constructed using environment knowledge with $\delta = 0.1$, and the corresponding trained actor is used to select actions. In the legend, *Irregular Transition* refers to state transitions that deviate from the permutation backbone; *Off-peak Observation* indicates observations that do not match the most likely ones given the current state; and *Decoding Error* denotes mismatches between the decoded and true states.

Two observations follow. First, **ALF** is more sensitive to noisy observations than **LOF**. For example, at timesteps 61 and 69, observations $B_3$ and $B_4$ occur at state 3, which are unlikely; these lead to decoding errors for **ALF** but not for **LOF**. Second, the learned policy is effective: the agent takes CW1 at state 2 and CCW1 at state 3 for most of the time, which leads to high return.

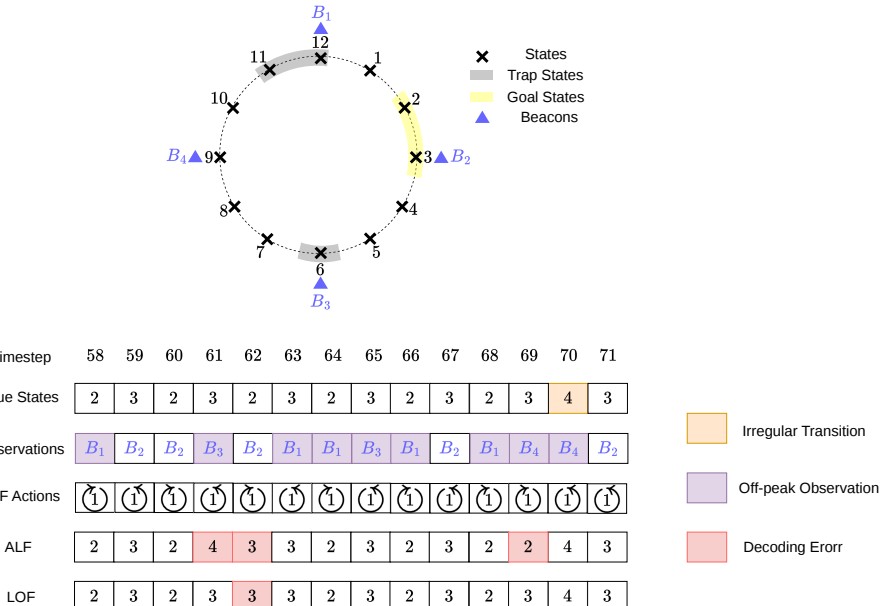

*Figure 6.* Sample trajectory of true states, estimated states, actions, and observations in the RingWorld environment.

## C. Proofs for Section 4

### C.1. Proof of Lemma 4.2

If $T$ is a permutation matrix, there are no transient states (i.e., $r = 0$), and Lemma 4.2 holds trivially as $T_{11}$ and $T_{21}$ are empty matrices. Otherwise, $T$ must have at least two identical columns, which implies the existence of at least one transient state: there must be a basis vector $u_i \in \{u_1, \ldots, u_N\}$ that does not appear in $T$. Consequently, no state can reach $x_i$, so $r > 0$, and the matrix $T$ is reducible.

According to the canonical form of a reducible matrix, $T$ can be expressed as (Meyer, 2023, (8.4.6); Puterman, 1990, (A.1)):

$$T = \left[ \begin{array}{c|c} T_{11} & 0 \\ \hline T_{21} & T_{22} \end{array} \right], \tag{55}$$

where the first $r$ states are transient, and the remaining $N - r$ are recurrent states. This matrix $T_{11} \in \mathbb{R}^{r \times r}$ is a lower-triangular block matrix whose diagonal blocks are either irreducible or $[0]_{1 \times 1}$. Additionally, $T_{21} \in \mathbb{R}^{(N-r) \times r}$ is non-zero, and $T_{22} \in \mathbb{R}^{(N-r) \times (N-r)}$ is a block diagonal matrix where each diagonal block is irreducible. This is a classical result for absorbing Markov chains. We now focus on its properties in the case of deterministic transitions.

We now prove, by contradiction, that all diagonal blocks in $T_{11}$ must be $[0]_{1 \times 1}$. If a diagonal block were irreducible, it would form an irreducible Markov chain isolated from other states, meaning no external state could enter it. Therefore, such states would not be transient, and they should appear in the bottom-right corner of $T$, which contradicts their placement in $T_{11}$.

Thus, $T$ can be written as:

$$\left[ \begin{array}{ccc|ccc} t_{11} & \cdots & 0 & & & \\ \vdots & \ddots & \vdots & & 0 & \\ t_{r1} & \cdots & t_{rr} & & & \\ \hline & & & P_{11} & \cdots & 0 \\ & T_{21} & & \vdots & \ddots & \vdots \\ & & & 0 & \cdots & P_{mm} \end{array} \right], \tag{56}$$

where $T_{21} \neq 0$, and the states $\{x_1, \ldots, x_r\}$ are transient. Moreover, all diagonal entries $t_{11}, \ldots, t_{rr}$ in $T_{11}$ are zero. The

blocks $P_{11}, \ldots, P_{mm}$ are irreducible, and since $T$ is deterministic, each $P_{ii}$ must be a permutation matrix. Consequently, the entire block $P$ is itself a permutation matrix.

### C.2. Proof of Theorem 4.4

We now consider the more general case where the number of transient states $r$ does not equal to 0. After removing the normalization in the action-independent version of (6) (we focus on a fixed-transition HMM), we obtain the **unnormalized optimal filter**:

$$
\begin{aligned}
\boldsymbol{\beta}_k &= \mathrm{diag}\{E^\top \boldsymbol{y}_k\} T \boldsymbol{\beta}_{k-1} \\
&= \mathrm{diag}\{E^\top \boldsymbol{y}_k\} T \cdots \mathrm{diag}\{E^\top \boldsymbol{y}_1\} T \pi_0.
\end{aligned}
\tag{57}
$$

We first show that

$$
\boldsymbol{\beta}_k = \begin{bmatrix} 0_r \\ \boldsymbol{\beta}_k(r+1) \\ \vdots \\ \boldsymbol{\beta}_k(N) \end{bmatrix} \triangleq \begin{bmatrix} 0_r \\ \widetilde{\boldsymbol{\beta}}_k \end{bmatrix},
\tag{58}
$$

for $k \geq \ell$, we use $0_r$ to denote zero vector with dimension $r$ to avoid confusion. We prove this by induction, if $\boldsymbol{\beta}_k$ admits the form (58), then

$$
\begin{aligned}
\boldsymbol{\beta}_{k+1} &= \mathrm{diag}\{E^\top \boldsymbol{y}_{k+1}\} T \boldsymbol{\beta}_k \\
&= \mathrm{diag}\{E^\top \boldsymbol{y}_{k+1}\} \begin{bmatrix} 0_r \\ P \widetilde{\boldsymbol{\beta}}_k \end{bmatrix} \\
&= \begin{bmatrix} 0_r \\ \widetilde{\boldsymbol{\beta}}_{k+1} \end{bmatrix}.
\end{aligned}
\tag{59}
$$

Therefore, what is left is to prove the base case where $\boldsymbol{\beta}_\ell$ admits the form (58), which is

$$
\begin{bmatrix} 0_r \\ \widetilde{\boldsymbol{\beta}}_\ell \end{bmatrix} = \underbrace{\mathrm{diag}\{E^\top \boldsymbol{y}_\ell\} T \cdots \mathrm{diag}\{E^\top \boldsymbol{y}_1\} T}_{\boldsymbol{U}_{1:\ell}} \pi_0.
\tag{60}
$$

Note that $T^\ell$ has:

$$
T^\ell = \begin{bmatrix} T_{11}^\ell & 0 \\ \cdot & P^\ell \end{bmatrix} \overset{(a)}{=} \begin{bmatrix} 0 & 0 \\ \cdot & P^\ell \end{bmatrix},
\tag{61}
$$

where $(a)$ is due to Lemma 4.2 and the property of the nilpotent matrix $T_{11}^\ell = 0$. Next, we show that $\boldsymbol{U}_{1:\ell}$ maintains the zeros in $T^\ell$. The matrix $\boldsymbol{U}_{1:\ell}$ can be expressed as:

$$
\boldsymbol{U}_{1:\ell} = \mathrm{diag}\{E^\top \boldsymbol{y}_\ell\} T \cdots \mathrm{diag}\{E^\top \boldsymbol{y}_1\} T.
\tag{62}
$$

Then we apply Lemma C.1 iteratively to prove that $\boldsymbol{U}_{1:\ell}$ maintains the zeros in $T^\ell$. This finishes the proof of (58).

It should be noted that the optimal logit can be obtained from $\boldsymbol{\beta}_k$ by:

$$
\boldsymbol{w}_k^o = \log \boldsymbol{\beta}_k.
\tag{63}
$$

Therefore, from (59), we have for the optimal logits $\boldsymbol{w}_{k+1}^o$ with $k \geq \ell$:

$$
\begin{aligned}
\boldsymbol{w}_{k+1}^o = \log(\boldsymbol{\beta}_{k+1}) &= \begin{bmatrix} -\infty \mathbb{1}_r \\ \log(\widetilde{\boldsymbol{\beta}}_{k+1}) \end{bmatrix} \\
&= \begin{bmatrix} I_r & 0 \\ 0 & P \end{bmatrix} \begin{bmatrix} -\infty \mathbb{1}_r \\ \log(\widetilde{\boldsymbol{\beta}}_k) \end{bmatrix} + \log(E^\top \boldsymbol{y}_{k+1}) \\
&= \begin{bmatrix} I_r & 0 \\ 0 & P \end{bmatrix} \boldsymbol{w}_k^o + \log(E^\top \boldsymbol{y}_{k+1}),
\end{aligned}
\tag{64}
$$

where we assume $-\infty \cdot 0 = 0$. Thus, to reproduce the optimal logits, we should have $\psi(k, \boldsymbol{y}_k, \cdots, \boldsymbol{y}_{k-\ell+1}) = \log(E^\top \boldsymbol{y}_k)$ when $k > \ell$. Also, since the evolution of the optimal logits $\boldsymbol{w}_k^o$ for $k > \ell$ relies on obtaining the exact $\log(\boldsymbol{\beta}_\ell)$ (i.e. $\boldsymbol{w}_\ell^o$), we need to reproduce $\boldsymbol{w}_\ell^o$ as well as $\boldsymbol{w}_k^o, 1 \le k < \ell$ precisely. Therefore, we need to design our $\psi(k, \boldsymbol{y}_k, \cdots, \boldsymbol{y}_{k-\ell+1})$ for $k \le \ell$. This is due to the fact that the evolution of $\boldsymbol{w}_k^o, 1 \le k \le \ell$ generally does not satisfy (64), and we need to design a $\psi$ to account for the errors induced by using the matrix $T'$.

The detailed design of $\psi$ relies on $\ell$. Instead of describing $\psi$ explicitly for all $\ell$, we only give a sample implementation of $\psi(k, y_2, y_1)$ for $\ell = 2$:

$$\psi(k, y_2, y_1) = \begin{cases} \log(T\pi_0) - T'w_0 + \log(E^\top y_2), & k = 1 \\ \log(T\mathrm{diag}\{E^\top y_1\}T\pi_0) - T'\log(\mathrm{diag}\{E^\top y_1\}T\pi_0) + \log(E^\top y_2), & k = 2 \\ \log(E^\top y_2), & k \ge 3 \end{cases}. \tag{65}$$

We can check that when $k > \ell = 2$, $\psi(k, y_2, y_1) = \log(E^\top y_2)$ which is the same as what we found in (64). Using this $\psi(k, y_2, y_1)$, our dynamics produces:

$$
\begin{aligned}
\boldsymbol{w}_1 &= T'\boldsymbol{w}_0 + \psi(1, \boldsymbol{y}_1, 0) \\
&= T'\boldsymbol{w}_0 + \log(T\pi_0) - T'\boldsymbol{w}_0 + \log(E^\top \boldsymbol{y}_1) \\
&= \log\big(\mathrm{diag}\{E^\top \boldsymbol{y}_1\} T\pi_0\big) \\
&= \boldsymbol{w}_1^o, \\
\boldsymbol{w}_2 &= T'\boldsymbol{w}_1 + \psi(2, \boldsymbol{y}_2, \boldsymbol{y}_1) \\
&= T'\log\big(\mathrm{diag}\{E^\top \boldsymbol{y}_1\} T\pi_0\big) + \log\big(T\,\mathrm{diag}\{E^\top \boldsymbol{y}_1\} T\pi_0\big) - T'\log\big(\mathrm{diag}\{E^\top \boldsymbol{y}_1\} T\pi_0\big) + \log(E^\top \boldsymbol{y}_2) \\
&= \log\big(\mathrm{diag}\{E^\top \boldsymbol{y}_2\} T\,\mathrm{diag}\{E^\top \boldsymbol{y}_1\} T\pi_0\big) \\
&= \boldsymbol{w}_2^o, \\
\boldsymbol{w}_3 &= T'\boldsymbol{w}_2 + \log(E^\top \boldsymbol{y}_3) \\
&= T'\boldsymbol{w}_2^o + \log(E^\top \boldsymbol{y}_3) \\
&= \boldsymbol{w}_3^o, \\
&\vdots
\end{aligned}
\tag{66}
$$

Note that $\psi$ for other values of $\ell$ can be constructed in similar ways.

### C.3. Supporting Lemma

**Lemma C.1.** *Assume all columns of matrix $B$ are standard basis vectors:*

$$B = \begin{bmatrix} u_{n_1} & \cdots & u_{n_N} \end{bmatrix}, \tag{67}$$

*we have:*

$$\mathrm{diag}\{c_1, \cdots, c_N\}B = B\,\mathrm{diag}\{c_{n_1}, \cdots, c_{n_N}\}. \tag{68}$$

*Proof.* The result follows from

$$
\begin{aligned}
&\mathrm{diag}\{c_1, \cdots, c_N\}B \\
&= \begin{bmatrix} \mathrm{diag}\{c_1, \cdots, c_N\}u_{n_1} & \cdots & \mathrm{diag}\{c_1, \cdots, c_N\}u_{n_N} \end{bmatrix} \\
&= \begin{bmatrix} c_{n_1}u_{n_1} & \cdots & c_{n_N}u_{n_N} \end{bmatrix} \\
&= \begin{bmatrix} u_{n_1} & \cdots & u_{n_N} \end{bmatrix}\mathrm{diag}\{c_{n_1}, \cdots, c_{n_N}\}.
\end{aligned}
\tag{69}
$$

$\square$

# D. Proof for Section 5

## D.1. Main Lemmas

**Lemma D.1.** *Under a nearly-deterministic transition matrix as defined in Definition 5.1, let $T_\varepsilon$ be defined as in (37). Then, under conditions (31)–(32), we have:*

$$\lim_{\varepsilon \to 0} \limsup_{k \to \infty} \sum_{x' \in \mathcal{X} \setminus \widetilde{\mathcal{X}}} \mathbb{P}\left[\boldsymbol{x}_k^\star = x'\right] = 0,$$

$$\lim_{\varepsilon \to 0} \limsup_{k \to \infty} \mathbb{P}\left[\text{IrJ in } [k - T_\varepsilon, k]\right] = 0. \tag{70}$$

*Proof.* From Lemma D.4, we have:

$$\limsup_{k \to \infty} \sum_{x' \in \mathcal{X} \setminus \widetilde{\mathcal{X}}} \mathbb{P}\left[\boldsymbol{x}_k^\star = x'\right] = O(\varepsilon), \tag{71}$$

which implies

$$\lim_{\varepsilon \to 0} \limsup_{k \to \infty} \sum_{x' \in \mathcal{X} \setminus \widetilde{\mathcal{X}}} \mathbb{P}\left[\boldsymbol{x}_k^\star = x'\right] = 0. \tag{72}$$

Similarly, from Lemma D.5, we have:

$$\limsup_{k \to \infty} \mathbb{P}\left[\text{IrJ in}[k - T_\varepsilon, k]\right] = O(\varepsilon T_\varepsilon) \tag{73}$$

As we can verify that our $T_\varepsilon$ in (37) satisfies the following two conditions according to (31) and (32):

$$\lim_{\varepsilon \to 0} \varepsilon T_\varepsilon = 0, \tag{74}$$

$$\lim_{\varepsilon \to 0} T_\varepsilon = +\infty, \tag{75}$$

we have:

$$\lim_{\varepsilon \to 0} \limsup_{k \to \infty} \mathbb{P}\left[\text{IrJ in } [k - T_\varepsilon, k]\right] = 0. \tag{76}$$

$\square$

**Lemma D.2.** *Under a nearly-deterministic transition matrix as defined in Definition 5.1, let $T_\varepsilon$ be defined as in (37). Then, under conditions (31)–(32), we have:*

$$\lim_{\varepsilon \to 0} \limsup_{k \to \infty} \mathbb{P}\left[\arg\max_{x \in \widetilde{\mathcal{X}}} \widetilde{\boldsymbol{w}}_k(x) \neq \boldsymbol{x}_k^\star \,\middle|\, \text{no IrJ in } I_k, \boldsymbol{x}_{k - h_\varepsilon M}^\star = x'\right] = 0, \ \forall x' \in \widetilde{\mathcal{X}} \tag{77}$$

*Proof.* We let $x' = x_1$ and prove

$$\lim_{\varepsilon \to 0} \limsup_{k \to \infty} \mathbb{P}\left[\arg\max_{x \in \widetilde{\mathcal{X}}} \widetilde{\boldsymbol{w}}_k(x) \neq \boldsymbol{x}_k^\star \,\middle|\, \text{no IrJ in } I_k, \boldsymbol{x}_{k - h_\varepsilon M}^\star = x_1\right] = 0, \tag{78}$$

the argument also holds for all $x' \in \widetilde{\mathcal{X}}$. Now we relabel the timestep $k - h_\varepsilon M$ as 0, then (77) can be rewritten as:

$$\mathbb{P}\left[\arg\max_{x \in \widetilde{\mathcal{X}}} \widetilde{\boldsymbol{w}}_{h_\varepsilon M}(x) \neq \boldsymbol{x}_{h_\varepsilon M}^\star \,\middle|\, \boldsymbol{x}_0^\star = x_1, \cdots, \boldsymbol{x}_{M-1}^\star = \sigma^{M-1}(x_1), \boldsymbol{x}_M^\star = x_1, \cdots, \boldsymbol{x}_{h_\varepsilon M}^\star = x_1\right]. \tag{79}$$

From the ALF dynamics (24), we have:

$$\widetilde{\boldsymbol{w}}_i = P(1 - \delta_\varepsilon)\widetilde{\boldsymbol{w}}_{i-1} + \delta_\varepsilon \log(\widetilde{E}^\top \boldsymbol{y}_i). \tag{80}$$

Now we define for $0 \leq i \leq h_\varepsilon M$,

$$\widehat{\boldsymbol{w}}_i \triangleq (P^\top)^i \widetilde{\boldsymbol{w}}_i = (P^\top)^{i \bmod M} \widetilde{\boldsymbol{w}}_i. \tag{81}$$

Therefore,

$$\underbrace{P^\top \widetilde{\boldsymbol{w}}_1}_{\widehat{\boldsymbol{w}}_1} = (1 - \delta_\varepsilon) \underbrace{\widetilde{\boldsymbol{w}}_0}_{\widehat{\boldsymbol{w}}_0} + \delta_\varepsilon P^\top \log(\widetilde{E}^\top \boldsymbol{y}_1),$$

$$\underbrace{(P^\top)^2 \widetilde{\boldsymbol{w}}_2}_{\widehat{\boldsymbol{w}}_2} = (1 - \delta_\varepsilon) \underbrace{P^\top \widetilde{\boldsymbol{w}}_1}_{\widehat{\boldsymbol{w}}_1} + \delta_\varepsilon (P^\top)^2 \log(\widetilde{E}^\top \boldsymbol{y}_2), \tag{82}$$

so on. Then, we have:

$$\widehat{\boldsymbol{w}}_i = (1 - \delta_\varepsilon) \widehat{\boldsymbol{w}}_{i-1} + \delta_\varepsilon (P^\top)^{i \bmod M} \log(\widetilde{E}^\top \boldsymbol{y}_i). \tag{83}$$

Now we define $\boldsymbol{o}_i \in \mathbb{R}^{N-r-1}$:

$$\boldsymbol{o}_i \triangleq \begin{bmatrix} \widehat{\boldsymbol{w}}_i(x_1) - \widehat{\boldsymbol{w}}_i(x_2) \\ \widehat{\boldsymbol{w}}_i(x_1) - \widehat{\boldsymbol{w}}_i(x_3) \\ \vdots \\ \widehat{\boldsymbol{w}}_i(x_1) - \widehat{\boldsymbol{w}}_i(x_{N-r}) \end{bmatrix} = \begin{bmatrix} \boldsymbol{o}_i(x_2) \\ \boldsymbol{o}_i(x_3) \\ \vdots \\ \boldsymbol{o}_i(x_{N-r}) \end{bmatrix}. \tag{84}$$

Here, $\widehat{\boldsymbol{w}}_i(x_1) = \widetilde{\boldsymbol{w}}_i(\sigma^{i \bmod M}(x_1))$ tracks the $\widetilde{\boldsymbol{w}}_i(x_i^\star)$ that corresponds to the true state $x_i^\star$ (see (79)). This can be verified by the following:

$$\widehat{\boldsymbol{w}}_i(x_1) = u_1^\top \widehat{\boldsymbol{w}}_i = u_1^\top (P^\top)^{i \bmod M} \widetilde{\boldsymbol{w}}_i$$
$$\overset{(a)}{=} u_{\sigma^{i \bmod M}(x_1)}^\top \widetilde{\boldsymbol{w}}_i = \widetilde{\boldsymbol{w}}_i(\sigma^{i \bmod M}(x_1)), \tag{85}$$

where $(a)$ is due to $u_1^\top (P^\top)^{i \bmod M} (P)^{i \bmod M} u_1 = 1$ and $(P)^{i \bmod M} u_1 = u_{\sigma^{i \bmod M}(x_1)}$. We can show that (79) becomes:

$$\mathbb{P}\left[\arg\max_{x \in \widetilde{\mathcal{X}}} \widetilde{\boldsymbol{w}}_{h_\varepsilon M}(x) \neq \boldsymbol{x}_{h_\varepsilon M}^\star \,\middle|\, \boldsymbol{x}_0^\star = x_1, \boldsymbol{x}_1^\star = \sigma(x_1), \cdots, \boldsymbol{x}_{h_\varepsilon M}^\star = x_1\right]$$
$$\overset{(a)}{\leq} \mathbb{P}\left[\exists x \in \widetilde{\mathcal{X}} \setminus \{x_1\}, \text{s.t. } \boldsymbol{o}_{h_\varepsilon M}(x) \leq 0 \,\middle|\, \boldsymbol{x}_0^\star = x_1, \boldsymbol{x}_1^\star = \sigma(x_1), \cdots, \boldsymbol{x}_{h_\varepsilon M}^\star = x_1\right]$$
$$\leq \sum_{x \in \widetilde{\mathcal{X}} \setminus \{x_1\}} \mathbb{P}\left[\boldsymbol{o}_{h_\varepsilon M}(x) \leq 0 \,\middle|\, \boldsymbol{x}_0^\star = x_1, \boldsymbol{x}_1^\star = \sigma(x_1), \cdots, \boldsymbol{x}_{h_\varepsilon M}^\star = x_1\right], \tag{86}$$

where $(a)$ is due to the fact that for any $x \in \widetilde{\mathcal{X}} \setminus \{x_1\}$,

$$\boldsymbol{o}_i(x) \leq 0 \Leftrightarrow \widehat{\boldsymbol{w}}_i(x_1) \leq \widehat{\boldsymbol{w}}_i(x) \Leftrightarrow \left[(P^{i \bmod M})^\top \widetilde{\boldsymbol{w}}_i(x)\right]_{x_1} \leq \left[(P^{i \bmod M})^\top \widetilde{\boldsymbol{w}}_i(x)\right]_x \tag{87}$$

and by substituting $i = h_\varepsilon M$, we have $P^{i \bmod M} = I$, which implies that

$$\boldsymbol{o}_{h_\varepsilon M}(x) \leq 0 \Leftrightarrow \widetilde{\boldsymbol{w}}_{h_\varepsilon M}(x_1) \leq \widetilde{\boldsymbol{w}}_{h_\varepsilon M}(x). \tag{88}$$

Now, for any $x \in \widetilde{\mathcal{X}} \setminus \{x_1\}$, we have:

$$\boldsymbol{o}_i(x) = (1 - \delta_\varepsilon) \boldsymbol{o}_{i-1}(x) + \delta_\varepsilon \left(\left[(P^\top)^{i \bmod M} \log(\widetilde{E}^\top \boldsymbol{y}_i)\right]_{x_1} - \left[(P^\top)^{i \bmod M} \log(\widetilde{E}^\top \boldsymbol{y}_i)\right]_x\right). \tag{89}$$

By recalling the definition of $E$ in (4), we can verify that:

$$\left[(P^\top)^{i \bmod M} \log(\widetilde{E}^\top \boldsymbol{y}_i)\right]_x = u_x^\top (P^\top)^{i \bmod M} \log(\widetilde{E}^\top \boldsymbol{y}_i) = \log(\mathbb{P}(\boldsymbol{y}_i \mid \sigma^{i \bmod M}(x))). \tag{90}$$

Therefore, (89) becomes:

$$\boldsymbol{o}_i(x) = (1 - \delta_\varepsilon)\boldsymbol{o}_{i-1}(x) + \delta_\varepsilon \log\left(\frac{\mathbb{P}(\boldsymbol{y}_i \mid \sigma^{i \bmod M}(x_1))}{\mathbb{P}(\boldsymbol{y}_i \mid \sigma^{i \bmod M}(x))}\right). \tag{91}$$

Here, we define:

$$\boldsymbol{g}_i(x) \triangleq \log\left(\frac{\mathbb{P}(\boldsymbol{y}_i \mid \sigma^{i \bmod M}(x_1))}{\mathbb{P}(\boldsymbol{y}_i \mid \sigma^{i \bmod M}(x))}\right), \tag{92}$$

then we have:

$$\boldsymbol{o}_{h_\varepsilon M}(x) = \underbrace{\delta_\varepsilon \sum_{m=0}^{h_\varepsilon M - 1} (1 - \delta_\varepsilon)^m \boldsymbol{g}_{h_\varepsilon M - m}(x)}_{\boldsymbol{o}'_{h_\varepsilon M}(x)} + (1 - \delta_\varepsilon)^{h_\varepsilon M}\boldsymbol{o}_0(x). \tag{93}$$

Now we continue from (86). In the following, since all probabilities and expectations are conditioned on the same event as in (86), we omit the conditional notation for clarity. We set

$$\gamma_\varepsilon \triangleq 1 - (1 - \delta_\varepsilon)^M, \tag{94}$$

and conclude for $x \in \widetilde{\mathcal{X}} \setminus \{x_1\}$:

$$
\begin{aligned}
&\mathbb{P}\left[\boldsymbol{o}_{h_\varepsilon M}(x) \leq 0\right] \\
&= \mathbb{P}\left[\boldsymbol{o}'_{h_\varepsilon M}(x) \leq -(1 - \delta_\varepsilon)^{h_\varepsilon M}\boldsymbol{o}_0(x)\right] \\
&\overset{(a)}{=} \mathbb{P}\left[-\frac{1}{\gamma_\varepsilon}\boldsymbol{o}'_{h_\varepsilon M}(x) \geq \frac{1}{\gamma_\varepsilon}(1 - \delta_\varepsilon)^{h_\varepsilon M}\boldsymbol{o}_0(x)\right] \\
&\overset{(b)}{\leq} \mathbb{P}\left[-\frac{1}{\gamma_\varepsilon}\boldsymbol{o}'_{h_\varepsilon M}(x) \geq -\frac{1}{\gamma_\varepsilon}(1 - \delta_\varepsilon)^{h_\varepsilon M}2W\right] \\
&\overset{(c)}{\leq} \frac{\mathbb{E}\left[\exp\left\{-\frac{1}{\gamma_\varepsilon}\boldsymbol{o}'_{h_\varepsilon M}(x)\right\}\right]}{\exp\left\{-\frac{1}{\gamma_\varepsilon}(1 - \delta_\varepsilon)^{h_\varepsilon M}2W\right\}} \\
&= \exp\left\{\frac{1}{\gamma_\varepsilon}\left[\gamma_\varepsilon\Lambda\left(-\frac{1}{\gamma_\varepsilon}; x\right) + 2(1 - \delta_\varepsilon)^{h_\varepsilon M}W\right]\right\},
\end{aligned} \tag{95}
$$

where $(a)$ is due to multiplying both sides by $-\frac{1}{\gamma_\varepsilon} < 0$. For $(b)$, we took advantage of the following:

$$
\begin{aligned}
\boldsymbol{o}_0(x) &= \widehat{\boldsymbol{w}}_0(x_1) - \widehat{\boldsymbol{w}}_0(x) \\
&= \widetilde{\boldsymbol{w}}_0(x_1) - \widetilde{\boldsymbol{w}}_0(x).
\end{aligned} \tag{96}
$$

Since $\|\widetilde{\boldsymbol{w}}_k\|_\infty \leq W, \forall k$ according to Lemma D.11 (recall that we labeled $k - h_\varepsilon M$ by 0 in (79), so our $\widetilde{\boldsymbol{w}}_0(x_1)$ is actually $\widetilde{\boldsymbol{w}}_{k-h_\varepsilon M}(x_1)$), we have

$$\boldsymbol{o}_0(x) \geq -2W, \tag{97}$$

which implies $(b)$. For $(c)$, we apply the Chernoff's bound. Here, we introduced the logarithmic moment generating function

of $\boldsymbol{o}'_{h_\varepsilon M}(x)$, which can be defined for $t \in \mathbb{R}$ as:

$$
\begin{aligned}
\Lambda(t;x) &\triangleq \log \mathbb{E}\left[e^{t\boldsymbol{o}'_{h_\varepsilon M}(x)}\right] \\
&= \log \mathbb{E}\left[\exp\left\{\sum_{m=0}^{h_\varepsilon M-1} \delta_\varepsilon(1-\delta_\varepsilon)^m t\boldsymbol{g}_{h_\varepsilon M-m}(x)\right\}\right] \\
&= \log \mathbb{E}\left[\exp\left\{\sum_{h=0}^{h_\varepsilon-1}\sum_{n=0}^{M-1} \delta_\varepsilon(1-\delta_\varepsilon)^{hM+n} t\boldsymbol{g}_{h_\varepsilon M-hM-n}(x)\right\}\right] \\
&\overset{(a)}{=} \sum_{h=0}^{h_\varepsilon-1}\sum_{n=0}^{M-1} \log \mathbb{E}\left[\exp\{\delta_\varepsilon(1-\delta_\varepsilon)^n(1-\delta_\varepsilon)^{hM} t\boldsymbol{g}_{h_\varepsilon M-hM-n}(x)\}\right] \\
&\overset{(b)}{=} \sum_{n=0}^{M-1}\sum_{h=0}^{h_\varepsilon-1} \Lambda_n\left(\delta_\varepsilon(1-\delta_\varepsilon)^n(1-\delta_\varepsilon)^{hM}t;x\right),
\end{aligned} \tag{98}
$$

where $(a)$ follows from the fact that, under the condition that the states of the Markov chain $\boldsymbol{x}_i^\star, 0 \le i \le h_\varepsilon M$ are fixed (see (86)), the observations $\boldsymbol{y}_1, \cdots, \boldsymbol{y}_{h_\varepsilon M}$ are independent and so are the $\boldsymbol{g}_i(x)$, since from (92), it can be seen that $\boldsymbol{g}_i(x)$ are functions of $\boldsymbol{y}_i$; $(b)$ introduced the following notation:

$$
\Lambda_n(t;x) \triangleq \log \mathbb{E}\left[e^{t\boldsymbol{g}_{h_\varepsilon M-n}(x)}\right]. \tag{99}
$$

Since $\boldsymbol{y}_{h_\varepsilon M-n}$ are discrete random variables, so are the $\boldsymbol{g}_{h_\varepsilon M-n}(x)$, therefore (99) and (98) are well defined for $t \in \mathbb{R}$. We also utilize the fact that every $M$ timesteps, $\boldsymbol{x}_i^\star$ are the same, as are the distributions of $\boldsymbol{y}_i$ and, correspondingly, $\boldsymbol{g}_i(x)$ (i.e., the distribution of $\boldsymbol{g}_{h_\varepsilon M-n}(x), \boldsymbol{g}_{h_\varepsilon M-M-n}(x), \cdots, \boldsymbol{g}_{M-n}(x)$ are identical).

We define:

$$
\begin{aligned}
\Gamma(x) &\triangleq \left\{n \in \{0, \cdots, M-1\}\big| E\left(u_{\sigma^{(h_\varepsilon M-n) \bmod M}(x)} - u_{\sigma^{(h_\varepsilon M-n) \bmod M}(x_1)}\right) \ne 0\right\} \\
&\overset{(a)}{=} \left\{n \in \{0, \cdots, M-1\}\big| E\left(u_{\sigma^{(-n) \bmod M}(x)} - u_{\sigma^{(-n) \bmod M}(x_1)}\right) \ne 0\right\} \\
&\overset{(b)}{=} \left\{n \in \{0, \cdots, M-1\}\big| \exists y \in \mathcal{Y}, \text{s.t. } \mathbb{P}\left(y\big|\sigma^{(-n) \bmod M}(x)\right) \ne \mathbb{P}\left(y\big|\sigma^{(-n) \bmod M}(x_1)\right)\right\},
\end{aligned} \tag{100}
$$

where $(a)$ is due to the cyclic property of the permutation $\sigma : \widetilde{\mathcal{X}} \to \widetilde{\mathcal{X}}$; $(b)$ is due to the definition of the emission matrix $E$. It can be proved that for all $x \in \widetilde{\mathcal{X}} \setminus x_1$, we have $|\Gamma(x)| \ge 1$ according to Lemma D.6. Furthermore, for those $n \in \{0, \cdots, M-1\} \setminus \Gamma(x)$, we have the following:

$$
\begin{aligned}
&\forall y \in \mathcal{Y}, \mathbb{P}\left(y\big|\sigma^{(-n) \bmod M}(x)\right) = \mathbb{P}\left(y\big|\sigma^{(-n) \bmod M}(x_1)\right) \\
\Rightarrow &\forall y \in \mathcal{Y}, \log\left(\frac{\mathbb{P}(y \mid \sigma^{(-n) \bmod M}(x_1))}{\mathbb{P}(y \mid \sigma^{(-n) \bmod M}(x))}\right) = 0 \\
\overset{(a)}{\Rightarrow} &\boldsymbol{g}_{h_\varepsilon M-n}(x) = 0 \quad \text{a.s.} \\
\Rightarrow &\Lambda_n(t;x) = 0, \forall t \in \mathbb{R},
\end{aligned} \tag{101}
$$

where $(a)$ follows from the definition of $\boldsymbol{g}_{h_\varepsilon M-n}$ in (92) and the cyclic property of permutation.

Therefore, $\Lambda(t;x)$ can be further simplified as follows:

$$
\Lambda(t;x) = \sum_{n \in \Gamma(x)}\sum_{h=0}^{h_\varepsilon-1} \Lambda_n\left(\delta_\varepsilon(1-\delta_\varepsilon)^n(1-\delta_\varepsilon)^{hM}t;x\right) \tag{102}
$$

For $n \in \Gamma(x)$, $\Lambda_n(t;x)$ has several interesting properties which are summarized in Lemma D.8, now we apply Lemma D.9

to prove an upper bound for the inner sum of (102). By setting $c = \delta_\varepsilon(1 - \delta_\varepsilon)^n$ and $\mu = 1 - (1 - \delta_\varepsilon)^M = \gamma_\varepsilon$, we have:

$$
\sum_{h=0}^{h_\varepsilon - 1} \Lambda_n \left( \delta_\varepsilon(1 - \delta_\varepsilon)^n (1 - \delta_\varepsilon)^{hM} t; x \right)
$$

$$
\leq \frac{1}{\gamma_\varepsilon} \left[ \int_{\delta_\varepsilon(1 - \delta_\varepsilon)^n (1 - \delta_\varepsilon)^{h_\varepsilon M} t}^{\delta_\varepsilon(1 - \delta_\varepsilon)^n t} \frac{\Lambda_n(\tau; x)}{\tau} d\tau + \frac{\gamma_\varepsilon}{2 - \gamma_\varepsilon} h_1(\delta_\varepsilon(1 - \delta_\varepsilon)^n t) \right]. \tag{103}
$$

By setting $t = -\frac{1}{\gamma_\varepsilon}$ and denoting

$$
\eta_\varepsilon \triangleq \frac{\delta_\varepsilon}{\gamma_\varepsilon}, \tag{104}
$$

we get:

$$
\sum_{h=0}^{h_\varepsilon - 1} \Lambda_n \left( -\eta_\varepsilon(1 - \delta_\varepsilon)^n (1 - \delta_\varepsilon)^{hM}; x \right)
$$

$$
\leq \frac{1}{\gamma_\varepsilon} \left[ \int_{-\eta_\varepsilon(1 - \delta_\varepsilon)^n (1 - \delta_\varepsilon)^{h_\varepsilon M}}^{-\eta_\varepsilon(1 - \delta_\varepsilon)^n} \frac{\Lambda_n(\tau; x)}{\tau} d\tau + \frac{\gamma_\varepsilon}{2 - \gamma_\varepsilon} h_1(-\eta_\varepsilon(1 - \delta_\varepsilon)^n) \right]
$$

$$
= \frac{1}{\gamma_\varepsilon} \left[ \int_0^{-\eta_\varepsilon(1 - \delta_\varepsilon)^n} \frac{\Lambda_n(\tau; x)}{\tau} d\tau - \int_0^{-\eta_\varepsilon(1 - \delta_\varepsilon)^n (1 - \delta_\varepsilon)^{h_\varepsilon M}} \frac{\Lambda_n(\tau; x)}{\tau} d\tau + \frac{\gamma_\varepsilon}{2 - \gamma_\varepsilon} h_1(-\eta_\varepsilon(1 - \delta_\varepsilon)^n) \right]
$$

$$
= \frac{1}{\gamma_\varepsilon} \left[ \int_0^{-\eta_\varepsilon(1 - \delta_\varepsilon)^n} \frac{\Lambda_n(\tau; x)}{\tau} d\tau + \int_{-\eta_\varepsilon(1 - \delta_\varepsilon)^n (1 - \delta_\varepsilon)^{h_\varepsilon M}}^0 \frac{\Lambda_n(\tau; x)}{\tau} d\tau + \frac{\gamma_\varepsilon}{2 - \gamma_\varepsilon} h_1(-\eta_\varepsilon(1 - \delta_\varepsilon)^n) \right]
$$

$$
\overset{(a)}{\leq} \frac{1}{\gamma_\varepsilon} \left[ \int_0^{-\eta_\varepsilon(1 - \delta_\varepsilon)^n} \frac{\Lambda_n(\tau; x)}{\tau} d\tau + \eta_\varepsilon(1 - \delta_\varepsilon)^n (1 - \delta_\varepsilon)^{h_\varepsilon M} d_n(x) + \frac{\gamma_\varepsilon}{2 - \gamma_\varepsilon} h_1(-\eta_\varepsilon(1 - \delta_\varepsilon)^n) \right], \tag{105}
$$

where $(a)$ follows from the fact that $\Lambda_n(\tau; x) \geq d_n(x)\tau$ for all $t \in \mathbb{R}$ in (2) of Lemma D.8.

Substituting the obtained results back into (102) and (95), we have:

$$
\mathbb{P}\left[ \boldsymbol{o}_{h_\varepsilon M}(x) \leq 0 \right]
$$

$$
\leq \exp\left\{ \frac{1}{\gamma_\varepsilon} \left[ \gamma_\varepsilon \Lambda\left( -\frac{1}{\gamma_\varepsilon}; x \right) + 2(1 - \delta_\varepsilon)^{h_\varepsilon M} W \right] \right\}
$$

$$
\overset{(a)}{\leq} \exp\left\{ \frac{1}{\gamma_\varepsilon} \left[ \sum_{n \in \Gamma(x)} \gamma_\varepsilon \sum_{h=0}^{h_\varepsilon - 1} \Lambda_n\left( -\eta_\varepsilon(1 - \delta_\varepsilon)^n (1 - \delta_\varepsilon)^{hM}; x \right) + 2(1 - \delta_\varepsilon)^{h_\varepsilon M} W \right] \right\}
$$

$$
\overset{(b)}{\leq} \exp\left\{ \frac{1}{\gamma_\varepsilon} \left[ \sum_{n \in \Gamma(x)} \left[ \int_0^{-\eta_\varepsilon(1 - \delta_\varepsilon)^n} \frac{\Lambda_n(\tau; x)}{\tau} d\tau + \eta_\varepsilon(1 - \delta_\varepsilon)^n (1 - \delta_\varepsilon)^{h_\varepsilon M} d_n(x) \right. \right. \right.
$$

$$
\left. \left. \left. + \frac{\gamma_\varepsilon}{2 - \gamma_\varepsilon} h_1(-\eta_\varepsilon(1 - \delta_\varepsilon)^n) \right] + 2(1 - \delta_\varepsilon)^{h_\varepsilon M} W \right] \right\}
$$

$$
= \exp\left\{ \frac{1}{\gamma_\varepsilon} \left[ \sum_{n \in \Gamma(x)} \int_0^{-\eta_\varepsilon(1 - \delta_\varepsilon)^n} \frac{\Lambda_n(\tau; x)}{\tau} d\tau + (1 - \delta_\varepsilon)^{h_\varepsilon M} \left( \sum_{n \in \Gamma(x)} \eta_\varepsilon(1 - \delta_\varepsilon)^n d_n(x) + 2W \right) \right. \right.
$$

$$
\left. \left. + \frac{\gamma_\varepsilon}{2 - \gamma_\varepsilon} \sum_{n \in \Gamma(x)} h_1(-\eta_\varepsilon(1 - \delta_\varepsilon)^n) \right] \right\}, \tag{106}
$$

where $(a)$ follows from (102); $(b)$ follows from (105). Now, from (86) and the original notation in (78), we have:

$$\limsup_{k\to\infty} \mathbb{P}\left[\arg\max_{x\in\widetilde{\mathcal{X}}} \widetilde{\boldsymbol{w}}_k(x) \neq \boldsymbol{x}_k^\star \middle| \text{no irregular jumps in} [k - h_\varepsilon M + 1, k], \boldsymbol{x}_k^\star = x_1\right]$$

$$\leq \sum_{x\in\widetilde{\mathcal{X}}\setminus\{x_1\}} \exp\left\{\frac{1}{\gamma_\varepsilon}\left[\sum_{n\in\Gamma(x)} \int_0^{-\eta_\varepsilon(1-\delta_\varepsilon)^n} \frac{\Lambda_n(\tau;x)}{\tau} d\tau + (1-\delta_\varepsilon)^{h_\varepsilon M}\left(\sum_{n\in\Gamma(x)} \eta_\varepsilon(1-\delta_\varepsilon)^n d_n(x) + 2W\right)\right.\right.$$

$$\left.\left. + \frac{\gamma_\varepsilon}{2-\gamma_\varepsilon}\sum_{n\in\Gamma(x)} h_1(-\eta_\varepsilon(1-\delta_\varepsilon)^n)\right]\right\}. \tag{107}$$

For $x \in \widetilde{\mathcal{X}} \setminus \{x_1\}$, by setting

$$A_\varepsilon(x) \triangleq \sum_{n\in\Gamma(x)} \int_0^{-\eta_\varepsilon(1-\delta_\varepsilon)^n} \frac{\Lambda_n(\tau;x)}{\tau} d\tau + (1-\delta_\varepsilon)^{h_\varepsilon M}\left(\sum_{n\in\Gamma(x)} \eta_\varepsilon(1-\delta_\varepsilon)^n d_n(x) + 2W\right)$$

$$+ \frac{\gamma_\varepsilon}{2-\gamma_\varepsilon}\sum_{n\in\Gamma(x)} h_1(-\eta_\varepsilon(1-\delta_\varepsilon)^n), \tag{108}$$

$$B_\varepsilon(x) \triangleq \exp\left\{\frac{1}{\gamma_\varepsilon} A_\varepsilon(x)\right\}, \tag{109}$$

we have:

$$\lim_{\varepsilon\to 0} A_\varepsilon(x)$$

$$= \lim_{\varepsilon\to 0} \sum_{n\in\Gamma(x)} \int_0^{-\eta_\varepsilon(1-\delta_\varepsilon)^n} \frac{\Lambda_n(\tau;x)}{\tau} d\tau + \lim_{\varepsilon\to 0}(1-\delta_\varepsilon)^{h_\varepsilon M}\left(\sum_{n\in\Gamma(x)} \eta_\varepsilon(1-\delta_\varepsilon)^n d_n(x) + 2W\right)$$

$$+ \lim_{\varepsilon\to 0}\frac{\gamma_\varepsilon}{2-\gamma_\varepsilon}\sum_{n\in\Gamma(x)} h_1(-\eta_\varepsilon(1-\delta_\varepsilon)^n)$$

$$\overset{(a)}{=} \sum_{n\in\Gamma(x)} \int_0^{-\frac{1}{M}} \frac{\Lambda_n(\tau;x)}{\tau} d\tau + \exp(-\alpha)\left(\sum_{n\in\Gamma(x)} \frac{1}{M} d_n(x) + 2W\right), \tag{110}$$

where $(a)$ follows from the facts that

$$\lim_{\varepsilon\to 0} \eta_\varepsilon(1-\delta_\varepsilon)^n = \frac{1}{M} \tag{111}$$

$$\lim_{\varepsilon\to 0}(1-\delta_\varepsilon)^{h_\varepsilon M} = \exp(-\alpha) \tag{112}$$

$$\lim_{\varepsilon\to 0}\frac{\gamma_\varepsilon}{2-\gamma_\varepsilon}\sum_{n\in\Gamma(x)} h_1(-\eta_\varepsilon(1-\delta_\varepsilon)^n) = 0. \tag{113}$$

We can prove (111) by noticing that

$$\lim_{\varepsilon\to 0} \eta_\varepsilon(1-\delta_\varepsilon)^n = \lim_{\delta_\varepsilon\to 0} \eta_\varepsilon(1-\delta_\varepsilon)^n = \lim_{\delta_\varepsilon\to 0} \frac{\delta_\varepsilon}{\gamma_\varepsilon} \overset{(a)}{=} \frac{1}{M}, \tag{114}$$

where $(a)$ follows from Lemma D.10. We can prove (112) by:

$$
\begin{aligned}
\lim_{\varepsilon \to 0}(1 - \delta_\varepsilon)^{h_\varepsilon M} &= \lim_{\delta_\varepsilon \to 0}(1 - \delta_\varepsilon)^{h_\varepsilon M} \\
&\overset{(a)}{=} \lim_{\delta_\varepsilon \to 0}(1 - \delta_\varepsilon)^{\lfloor \frac{T_\varepsilon}{M} \rfloor M} \\
&\overset{(b)}{=} \lim_{\delta_\varepsilon \to 0}(1 - \delta_\varepsilon)^{T_\varepsilon} \\
&= \lim_{\delta_\varepsilon \to 0}(1 - \delta_\varepsilon)^{\frac{\alpha}{\delta_\varepsilon}} \\
&= \exp(-\alpha),
\end{aligned}
\tag{115}
$$

where $(a)$ follows from the definition of $h_\varepsilon$ in (38); For $(b)$, we have

$$
(\frac{T_\varepsilon}{M} - 1)M \le \lfloor \frac{T_\varepsilon}{M} \rfloor M \le T_\varepsilon
$$
$$
\Rightarrow (1 - \delta_\varepsilon)^{T_\varepsilon} \le (1 - \delta_\varepsilon)^{\lfloor \frac{T_\varepsilon}{M} \rfloor M} \le (1 - \delta_\varepsilon)^{T_\varepsilon - M},
\tag{116}
$$

and since $M$ does not depend on $\delta_\varepsilon$, the limit on the right hand side can be established as:

$$
\lim_{\delta_\varepsilon \to 0}(1 - \delta_\varepsilon)^{T_\varepsilon - M} = \lim_{\delta_\varepsilon \to 0}(1 - \delta_\varepsilon)^{T_\varepsilon} \lim_{\delta_\varepsilon \to 0}(1 - \delta_\varepsilon)^{-M} = \lim_{\delta_\varepsilon \to 0}(1 - \delta_\varepsilon)^{T_\varepsilon}.
\tag{117}
$$

Equation (113) can be proved by:

$$
\begin{aligned}
\lim_{\varepsilon \to 0} \frac{\gamma_\varepsilon}{2 - \gamma_\varepsilon} \sum_{n \in \Gamma(x)} h_1(-\eta_\varepsilon(1 - \delta_\varepsilon)^n) &= \lim_{\varepsilon \to 0} \frac{\gamma_\varepsilon}{2 - \gamma_\varepsilon} \sum_{n \in \Gamma(x)} \lim_{\varepsilon \to 0} h_1(-\eta_\varepsilon(1 - \delta_\varepsilon)^n) \\
&\overset{(a)}{=} 0 \cdot h_1(-\frac{1}{M}) \\
&= 0,
\end{aligned}
\tag{118}
$$

where $(a)$ follows from Lemma D.10 and (111).

Now, we consider the following function for $t \in \mathbb{R}$:

$$
r(t) = \int_0^t \sum_{n \in \Gamma(x)} \frac{\Lambda_n(\tau; x)}{\tau} d\tau.
\tag{119}
$$

This $r(t)$ is strictly convex since $r''(t) > 0, \forall t \in \mathbb{R}$ according to (5) of Lemma D.8. We have $r(0) = 0$ and

$$
r'(0) = \sum_{n \in \Gamma(x)} \frac{\Lambda_n(\tau; x)}{\tau} \bigg|_{\tau=0} \overset{(a)}{=} \sum_{n \in \Gamma(x)} d_n(x) > 0,
\tag{120}
$$

where $(a)$ follows from (2) of Lemma D.8. Also, from (2) of Lemma D.8, $r(t)$ reaches its minimum $r(-1) < 0$ at $t = -1$:

$$
r'(-1) = \sum_{n \in \Gamma(x)} \frac{\Lambda_n(-1; x)}{-1} = 0.
\tag{121}
$$

Therefore, $r(t)$ increases monotonically from $r(-1) < 0$ to $r(0) = 0$ for $t \in [-1, 0]$, and it must have $r(-\frac{1}{M}) < 0$. Hence, the first term in (110) is strictly smaller than 0. Now, if we set

$$
\alpha \ge \alpha_1(x) = \log(\sum_{n \in \Gamma(x)} \frac{1}{M} d_n(x) + 2W) - \log(-\frac{1}{2} \int_0^{-\frac{1}{M}} \sum_{n \in \Gamma(x)} \frac{\Lambda_n(\tau; x)}{\tau} d\tau),
\tag{122}
$$

and substitute it back into (110), we obtain:

$$
\lim_{\varepsilon \to 0} A_\varepsilon(x) \le \frac{1}{2} \int_0^{-\frac{1}{M}} \sum_{n \in \Gamma(x)} \frac{\Lambda_n(\tau; x)}{\tau} d\tau < 0.
\tag{123}
$$

Now letting $\alpha \geq \max_{x \in \mathcal{X} \setminus \{x_1\}} \alpha_1(x)$, we conclude the proof:

$$
\lim_{\varepsilon \to 0} \limsup_{k \to \infty} \mathbb{P}\left[\arg\max_{x \in \widetilde{\mathcal{X}}} \widetilde{\boldsymbol{w}}_k(x) \neq \boldsymbol{x}_k^\star \middle| \text{ no irregular jumps in} [k - h_\varepsilon M + 1, k], \boldsymbol{x}_k^\star = x_1\right]
$$
$$
\overset{(a)}{\leq} \sum_{x \in \widetilde{\mathcal{X}} \setminus \{x_1\}} \lim_{\varepsilon \to 0} B_\varepsilon(x)
$$
$$
= \sum_{x \in \widetilde{\mathcal{X}} \setminus \{x_1\}} \exp\{\lim_{\varepsilon \to 0} \frac{1}{\gamma_\varepsilon} \lim_{\varepsilon \to 0} A_\varepsilon(x)\}
$$
$$
= \sum_{x \in \widetilde{\mathcal{X}} \setminus \{x_1\}} \exp\{-\infty\}
$$
$$
= 0, \tag{124}
$$

where $(a)$ follows from (107).

$\square$

**Lemma D.3.** *Consider a nearly-deterministic transition matrix $T$ under Assumptions 5.3 and 5.6. If we choose*

$$
\delta_\varepsilon = \frac{\lambda}{\log(1/\varepsilon)}, \quad \text{with } \lambda \in (0, \xi), \tag{125}
$$

*where $\xi > 0$ is a constant defined in (145) to ensure $\delta_\varepsilon$ small enough, then it holds that*

$$
\limsup_{k \to \infty} p_k \leq \kappa \, \varepsilon \log\frac{1}{\varepsilon} + o(\varepsilon \log\frac{1}{\varepsilon}). \tag{126}
$$

*The resulting decay rate matches that of Bayes-optimal decoder (Khasminskii & Zeitouni, 1996), in the sense that the long-term error is also dominated by an $\varepsilon \log(1/\varepsilon)$ term. The constant factor $\kappa > 0$ is specified in (148).*

*Proof.* From (40), we obtain that

$$
\limsup_{k \to \infty} p_k \leq \sum_{x' \in \widetilde{\mathcal{X}}} \limsup_{k \to \infty} \mathbb{P}\left[\arg\max_{x \in \widetilde{\mathcal{X}}} \widetilde{\boldsymbol{w}}_k(x) \neq \boldsymbol{x}_k^\star \middle| \text{no IrJ in } I_k, \boldsymbol{x}_{k-h_\varepsilon M}^\star = x'\right]
$$
$$
+ \limsup_{k \to \infty} \sum_{x' \in \mathcal{X} \setminus \widetilde{\mathcal{X}}} \mathbb{P}\left[\boldsymbol{x}_{k-h_\varepsilon M}^\star = x'\right] + \limsup_{k \to \infty} \mathbb{P}\left[\text{IrJ in } [k - T_\varepsilon, k]\right]. \tag{127}
$$

By Lemma D.4, we have:

$$
\limsup_{k \to \infty} \sum_{x' \in \mathcal{X} \setminus \widetilde{\mathcal{X}}} \mathbb{P}\left[\boldsymbol{x}_{k-h_\varepsilon M}^\star = x'\right] = O(\varepsilon) = o(\varepsilon \log(\frac{1}{\varepsilon})). \tag{128}
$$

Moreover, from Lemma D.5, we have:

$$
\limsup_{k \to \infty} \mathbb{P}\left[\text{IrJ in } [k - T_\varepsilon, k]\right] \leq (T_\varepsilon + 1)\varepsilon(-q_{\min}) = -q_{\min}T_\varepsilon \varepsilon + (-q_{\min})\varepsilon
$$
$$
\overset{(a)}{=} \frac{-q_{\min}\alpha}{\lambda} \varepsilon \log(\frac{1}{\varepsilon}) + o(\varepsilon \log(\frac{1}{\varepsilon})), \tag{129}
$$

where $(a)$ follows from the definitions of $T_\varepsilon$ and $\delta_\varepsilon$ in (37) and (125), which imply

$$
T_\varepsilon = \frac{\alpha}{\delta_\varepsilon} = \frac{\alpha}{\lambda} \log(\frac{1}{\varepsilon}). \tag{130}
$$

Here, $q_{\min}$ is related to matrix $Q$ and defined in (159). Now, we only need to focus on the first term. In the following, we let $x' = x_1$ and prove that

$$\limsup_{k \to \infty} \mathbb{P}\left[\arg\max_{x \in \widetilde{\mathcal{X}}} \widetilde{\boldsymbol{w}}_k(x) \neq \boldsymbol{x}_k^\star \,\middle|\, \text{no IrJ in } I_k, \boldsymbol{x}_{k-h_\varepsilon M}^\star = x_1\right] \leq o(\varepsilon \log(\frac{1}{\varepsilon})), \tag{131}$$

and the same arguments apply for all $x' \in \widetilde{\mathcal{X}}$ without loss of generality. Recall from (107), we have

$$\limsup_{k \to \infty} \mathbb{P}\left[\arg\max_{x \in \widetilde{\mathcal{X}}} \widetilde{\boldsymbol{w}}_k(x) \neq \boldsymbol{x}_k^\star \,\middle|\, \text{no irregular jumps in} [k - h_\varepsilon M + 1, k], \boldsymbol{x}_k^\star = x_1\right]$$

$$\leq \sum_{x \in \widetilde{\mathcal{X}} \setminus \{x_1\}} \exp\left\{\frac{1}{\gamma_\varepsilon}\left[\sum_{n \in \Gamma(x)} \int_0^{-\eta_\varepsilon(1-\delta_\varepsilon)^n} \frac{\Lambda_n(\tau; x)}{\tau} d\tau + (1-\delta_\varepsilon)^{h_\varepsilon M}\left(\sum_{n \in \Gamma(x)} \eta_\varepsilon(1-\delta_\varepsilon)^n d_n(x) + 2W\right)\right.\right.$$

$$\left.\left. + \frac{\gamma_\varepsilon}{2-\gamma_\varepsilon} \sum_{n \in \Gamma(x)} h_1(-\eta_\varepsilon(1-\delta_\varepsilon)^n)\right]\right\}. \tag{132}$$

We define the function

$$f(\varepsilon; x) \triangleq \exp\left\{\frac{1}{\gamma_\varepsilon}\left[\sum_{n \in \Gamma(x)} \int_0^{-\eta_\varepsilon(1-\delta_\varepsilon)^n} \frac{\Lambda_n(\tau; x)}{\tau} d\tau + (1-\delta_\varepsilon)^{h_\varepsilon M}\left(\sum_{n \in \Gamma(x)} \eta_\varepsilon(1-\delta_\varepsilon)^n d_n(x) + 2W\right)\right.\right.$$

$$\left.\left. + \frac{\gamma_\varepsilon}{2-\gamma_\varepsilon} \sum_{n \in \Gamma(x)} h_1(-\eta_\varepsilon(1-\delta_\varepsilon)^n)\right]\right\}, \tag{133}$$

and consider the ratio

$$\frac{f(\varepsilon; x)}{\varepsilon \log(\frac{1}{\varepsilon})} = \frac{1}{\log(\frac{1}{\varepsilon})} \exp\left\{\frac{1}{\gamma_\varepsilon}\left[\sum_{n \in \Gamma(x)} \int_0^{-\eta_\varepsilon(1-\delta_\varepsilon)^n} \frac{\Lambda_n(\tau; x)}{\tau} d\tau + (1-\delta_\varepsilon)^{h_\varepsilon M}\left(\sum_{n \in \Gamma(x)} \eta_\varepsilon(1-\delta_\varepsilon)^n d_n(x) + 2W\right)\right.\right.$$

$$\left.\left. + \frac{\gamma_\varepsilon}{2-\gamma_\varepsilon} \sum_{n \in \Gamma(x)} h_1(-\eta_\varepsilon(1-\delta_\varepsilon)^n)\right] + \log(\frac{1}{\varepsilon})\right\}. \tag{134}$$

Based on $\delta_\varepsilon = \lambda / \log(1/\varepsilon)$ in (125), we have

$$\log(\frac{1}{\varepsilon}) = \lambda \frac{1}{\delta_\varepsilon} = \lambda M \frac{1}{M \delta_\varepsilon} \overset{(a)}{\leq} \lambda M \frac{1}{1-(1-\delta_\varepsilon)^M} \overset{(b)}{=} \lambda M \frac{1}{\gamma_\varepsilon}, \tag{135}$$

where $(a)$ follows from the inequality $(1+x)^M \geq 1 + Mx, \forall x \geq -1$ and by setting $x = -\delta_\varepsilon$; $(b)$ follows from the definition of $\gamma_\varepsilon$ in (94).

Taking (135) into (134), we have:

$$\frac{f(\varepsilon; x)}{\varepsilon \log(\frac{1}{\varepsilon})} \leq \frac{1}{\log(\frac{1}{\varepsilon})} \exp\left\{\frac{1}{\gamma_\varepsilon}\left[\sum_{n \in \Gamma(x)} \int_0^{-\eta_\varepsilon(1-\delta_\varepsilon)^n} \frac{\Lambda_n(\tau; x)}{\tau} d\tau + (1-\delta_\varepsilon)^{h_\varepsilon M}\left(\sum_{n \in \Gamma(x)} \eta_\varepsilon(1-\delta_\varepsilon)^n d_n(x) + 2W\right)\right.\right.$$

$$\left.\left. + \frac{\gamma_\varepsilon}{2-\gamma_\varepsilon} \sum_{n \in \Gamma(x)} h_1(-\eta_\varepsilon(1-\delta_\varepsilon)^n) + \lambda M\right]\right\}. \tag{136}$$

We define

$$C_\varepsilon(x) = \sum_{n \in \Gamma(x)} \int_0^{-\eta_\varepsilon(1-\delta_\varepsilon)^n} \frac{\Lambda_n(\tau; x)}{\tau} d\tau + (1-\delta_\varepsilon)^{h_\varepsilon M}\left(\sum_{n \in \Gamma(x)} \eta_\varepsilon(1-\delta_\varepsilon)^n d_n(x) + 2W\right)$$

$$+ \frac{\gamma_\varepsilon}{2-\gamma_\varepsilon} \sum_{n \in \Gamma(x)} h_1(-\eta_\varepsilon(1-\delta_\varepsilon)^n) + \lambda M. \tag{137}$$

Then we have:

$$\lim_{\varepsilon \to 0} C_\varepsilon(x) = \lim_{\varepsilon \to 0} \sum_{n \in \Gamma(x)} \int_0^{-\eta_\varepsilon (1-\delta_\varepsilon)^n} \frac{\Lambda_n(\tau; x)}{\tau} d\tau + \lim_{\varepsilon \to 0} (1-\delta_\varepsilon)^{h_\varepsilon M} \left( \sum_{n \in \Gamma(x)} \eta_\varepsilon (1-\delta_\varepsilon)^n d_n(x) + 2W \right)$$

$$+ \lim_{\varepsilon \to 0} \frac{\gamma_\varepsilon}{2 - \gamma_\varepsilon} \sum_{n \in \Gamma(x)} h_1(-\eta_\varepsilon (1-\delta_\varepsilon)^n) + \lambda M$$

$$\overset{(a)}{=} \sum_{n \in \Gamma(x)} \int_0^{-\frac{1}{M}} \frac{\Lambda_n(\tau; x)}{\tau} d\tau + \lambda M + \exp(-\alpha) \left( \sum_{n \in \Gamma(x)} \frac{1}{M} d_n(x) + 2W \right), \tag{138}$$

where $(a)$ is due to (111), (112), and (113) proved previously. Given that the first term is smaller than 0 (see (119)-(121)), if $\lambda$ satisfies

$$\lambda < -\frac{1}{M} \sum_{n \in \Gamma(x)} \int_0^{-\frac{1}{M}} \frac{\Lambda_n(\tau; x)}{\tau} d\tau, \tag{139}$$

then there always exists a large enough $\alpha$ such that

$$\lim_{\varepsilon \to 0} C_\varepsilon(x) < 0. \tag{140}$$

Now we have:

$$\lim_{\varepsilon \to 0} \frac{f(\varepsilon; x)}{\varepsilon \log(\frac{1}{\varepsilon})} \le \lim_{\varepsilon \to 0} \frac{1}{\log(\frac{1}{\varepsilon})} \exp \left\{ \lim_{\varepsilon \to 0} \frac{1}{\gamma_\varepsilon} \lim_{\varepsilon \to 0} C_\varepsilon(x) \right\} = 0 \cdot \exp(-\infty) = 0, \tag{141}$$

and we can conclude that

$$\lim_{\varepsilon \to 0} \frac{f(\varepsilon; x)}{\varepsilon \log(\frac{1}{\varepsilon})} = 0. \tag{142}$$

From (132), we have:

$$\limsup_{k \to \infty} \mathbb{P} \left[ \arg\max_{x \in \widetilde{\mathcal{X}}} \widetilde{\boldsymbol{w}}_k(x) \ne \boldsymbol{x}_k^\star \,\middle|\, \text{no irregular jumps in} [k - h_\varepsilon M + 1, k], \boldsymbol{x}_k^\star = x_1 \right]$$

$$\le \sum_{x \in \widetilde{\mathcal{X}} \backslash \{x_1\}} f(\varepsilon; x)$$

$$\overset{(a)}{=} o(\varepsilon \log(\frac{1}{\varepsilon})). \tag{143}$$

where $(a)$ follows from the choice of

$$\lambda < \xi(x_1) \triangleq \min_{x \in \widetilde{\mathcal{X}} \backslash \{x_1\}} -\frac{1}{M} \sum_{n \in \Gamma(x)} \int_0^{-\frac{1}{M}} \frac{\Lambda_n(\tau; x)}{\tau} d\tau. \tag{144}$$

Now, going back to (127), the above derivation considers the case $x' = x_1$ without loss of generality. Similar steps can be carried out for all $x' \in \widetilde{\mathcal{X}}$, yielding an overall choice of

$$\lambda < \xi \triangleq \min_{x' \in \widetilde{\mathcal{X}}} \{\xi(x')\} \tag{145}$$

that ensures

$$\sum_{x' \in \widetilde{\mathcal{X}}} \limsup_{k \to \infty} \mathbb{P} \left[ \arg\max_{x \in \widetilde{\mathcal{X}}} \widetilde{\boldsymbol{w}}_k(x) \ne \boldsymbol{x}_k^\star \,\middle|\, \text{no IrJ in } I_k, \boldsymbol{x}_{k-h_\varepsilon M}^\star = x' \right] \le o\left(\varepsilon \log \frac{1}{\varepsilon}\right). \tag{146}$$

Combining (127), (128), (129), and (143), we obtain:

$$\limsup_{k \to \infty} p_k \le \frac{-q_{\min}\alpha}{\lambda}\varepsilon \log(\frac{1}{\varepsilon}) + o(\varepsilon \log(\frac{1}{\varepsilon}))$$

$$\overset{(a)}{=} \kappa\varepsilon \log(\frac{1}{\varepsilon}) + o(\varepsilon \log(\frac{1}{\varepsilon})), \tag{147}$$

where in $(a)$ we define

$$\kappa \triangleq \frac{-q_{\min}\alpha}{\lambda}. \tag{148}$$

Now we complete the proof. $\qquad\square$

### D.2. Discussion on $\xi$

The constant $\xi$ defined in (145) plays an important role in this state-decoding problem. Here, we provide a more detailed discussion of this constant.

As shown in (144)–(145), $\xi$ depends on the permutation backbone $P$ and the emission model $E$ through the quantities $M$, the order of the permutation matrix $P$, the set $\Gamma(x)$, and $\Lambda_n$, the moment generating function of the log-likelihood ratios. This $\Lambda_n$ characterizes the intrinsic difficulty of the inference task. In particular, the emission matrix $E$ directly influences $\Lambda_n$: when the likelihoods under different states (i.e., the columns of $E$) are more distinguishable, $\xi$ becomes larger, allowing for a larger choice of $\lambda$ in (34). When the emission model is known, $\xi$ can be computed analytically from (145). When the emission model is unknown or too complex, $\xi$ can instead be estimated empirically from data. The parameter $\xi$ is related to the notion of the *error exponent* in the social learning literature; see, e.g., Khammassi et al. (2025).

Now we provide an example of $\xi$ computation corresponding to the model in Section 6.1: We consider a two-state HMM without transient states ($r = 0$), thus we have $\mathcal{X} = \widetilde{\mathcal{X}} = \{x_1, x_2\}$. We have the permutation order $M = 2$. Due to the symmetry, we have

$$\xi = \xi(x_1)$$

$$= \min_{x \in \widetilde{\mathcal{X}} \setminus \{x_1\}} -\frac{1}{2} \sum_{n \in \Gamma(x)} \int_0^{-\frac{1}{2}} \frac{\Lambda_n(\tau; x)}{\tau} d\tau$$

$$\overset{(a)}{=} \sum_{n=0}^{1} -\frac{1}{2} \int_0^{-\frac{1}{2}} \frac{\Lambda_n(\tau; x_2)}{\tau} d\tau$$

$$\overset{(b)}{=} -\int_0^{-\frac{1}{2}} \frac{\Lambda_0(\tau; x_2)}{\tau} d\tau \tag{149}$$

where $(a)$ follows from the fact that $\widetilde{\mathcal{X}} \setminus \{x_1\} = \{x_2\}$, $\Gamma(x_2) = \{0, 1\}$ since the two columns in $E$ are not identical; $(b)$ is due to the symmetry of the likelihood model determined by $E$. Then, following the definition of $\Lambda_n$, we have

$$\xi = -\int_0^{-\frac{1}{2}} \frac{\log\big(0.9\, e^{t \log(0.9/0.1)} + 0.1\, e^{t \log(0.1/0.9)}\big)}{t} dt \approx 0.7206. \tag{150}$$

### D.3. Supporting Lemmas

**Lemma D.4.** *Consider a nearly-deterministic transition matrix $T$ with $D$ in (22), let $\ell$ be the nilpotent index of $D_{22}$. Then, we have for $k \ge \ell$:*

$$\mathbb{P}[\boldsymbol{x}_k^\star = x] = O(\varepsilon), \quad \forall x \in \mathcal{X} \setminus \widetilde{\mathcal{X}}. \tag{151}$$

*Proof.* In the following, we assume $D$ has the from in (22). The nearly-deterministic transition matrix $T$ can be formed as:

$$T = \left[ \begin{array}{c|c} P + \varepsilon Q_{11} & D_{12} + \varepsilon Q_{12} \\ \hline \varepsilon Q_{21} & D_{22} + \varepsilon Q_{22} \end{array} \right], \tag{152}$$

where the blocks $Q_{11} \in \mathbb{R}^{r \times r}, Q_{12} \in \mathbb{R}^{r \times N-r}, Q_{21} \in \mathbb{R}^{N-r \times r}$, and $Q_{22} \in \mathbb{R}^{N-r \times N-r}$ form the matrix $Q$. Then, we have:

$$T^2 = \left[ \begin{array}{c|c} \cdot & \cdot \\ \hline (D_{22} + \varepsilon Q_{22})\varepsilon Q_{21} + \varepsilon Q_{21}(P + \varepsilon Q_{11}) & (D_{22} + \varepsilon Q_{22})^2 + \varepsilon Q_{21}(D_{12} + \varepsilon Q_{12}) \end{array} \right]$$

$$= \left[ \begin{array}{c|c} \cdot & \cdot \\ \hline O(\varepsilon) & D_{22}^2 + O(\varepsilon) \end{array} \right]. \tag{153}$$

Similarly, for $T^\ell$ we can get:

$$T^\ell = \left[ \begin{array}{c|c} \cdot & \cdot \\ \hline O(\varepsilon) & D_{22}^\ell + O(\varepsilon) \end{array} \right]$$

$$\overset{(a)}{=} \left[ \begin{array}{c|c} \cdot & \cdot \\ \hline O(\varepsilon) & O(\varepsilon) \end{array} \right]. \tag{154}$$

where $(a)$ follows from the fact that $D_{22}$ is a nilpotent matrix with index $\ell$. For $k \geq \ell$, we always have:

$$T^k = \left[ \begin{array}{c|c} \cdot & \cdot \\ \hline O(\varepsilon) & O(\varepsilon) \end{array} \right]. \tag{155}$$

Therefore, in conclusion, for $k \geq \ell$ and $\forall x \in \mathcal{X} \setminus \widetilde{\mathcal{X}}$, we have:

$$\mathbb{P}\left[\boldsymbol{x}_k^\star = x\right] = \pi_k(x) = \left[(T)^k \pi_0\right]_x = O(\varepsilon). \tag{156}$$

$\square$

**Lemma D.5.** *Under a nearly-deterministic transition matrix as defined in Definition 5.1, we have*

$$\mathbb{P}\left[\text{one or more irregular jumps in } [k - T_\varepsilon, k]\right] = O(\varepsilon T_\varepsilon). \tag{157}$$

$\square$

*Proof.* Since

$$\mathbb{P}\left[\text{one or more irregular jumps in } [k - T_\varepsilon, k]\right]$$
$$= 1 - \mathbb{P}\left[\text{no irregular jumps in } [k - T_\varepsilon, k]\right]. \tag{158}$$

The probability of not having an irregular jump at a single step $k$, given $\boldsymbol{x}_{k-1}^\star = x_j$, is $1 + \varepsilon \, q_{n_j j}$. Since $q_{n_j j} \leq 0$, define

$$q_{\max} \triangleq \max_j q_{n_j j}, \qquad q_{\min} \triangleq \min_j q_{n_j j}. \tag{159}$$

The probability of not having irregular jumps at any step lies between $1 + \varepsilon q_{\min}$ and $1 + \varepsilon q_{\max}$. Therefore,

$$(1 + \varepsilon q_{\min})^{\lceil T_\varepsilon \rceil} \leq \mathbb{P}[\text{no irregular jumps in } [k - \lceil T_\varepsilon \rceil, k]]$$
$$\leq \mathbb{P}[\text{no irregular jumps in } [k - T_\varepsilon, k]]$$
$$\leq \mathbb{P}[\text{no irregular jumps in } [k - \lfloor T_\varepsilon \rfloor, k]]$$
$$\leq (1 + \varepsilon q_{\max})^{\lfloor T_\varepsilon \rfloor}. \tag{160}$$

This implies that

$$\mathbb{P}\left[\text{one or more irregular jumps in } [k - T_\varepsilon, k]\right] \leq 1 - (1 + \varepsilon q_{\min})^{\lceil T_\varepsilon \rceil}. \tag{161}$$

We use Bernoulli's inequality, which states that for any real number $x \geq -1$ and any integer $n \geq 0$:

$$(1 + x)^n \geq 1 + nx. \tag{162}$$

Now let $x = \varepsilon q_{\min}, n = \lceil T_\varepsilon \rceil$, if $\varepsilon$ is small enough such that

$$\varepsilon \leq -\frac{1}{q_{\min}}, \tag{163}$$

we have:

$$(1 + \varepsilon q_{\min})^{\lfloor T_\varepsilon \rfloor} \geq 1 + \lceil T_\varepsilon \rceil \varepsilon q_{\min}. \tag{164}$$

Now we have the upper bound:

$$\begin{aligned}
& \mathbb{P}\left[\text{one or more irregular jumps in } [k - T_\varepsilon, k]\right] \\
& \leq 1 - (1 + \varepsilon q_{\min})^{\lceil T_\varepsilon \rceil} \\
& \leq \lceil T_\varepsilon \rceil \varepsilon (-q_{\min}) \\
& \leq (T_\varepsilon + 1) \varepsilon (-q_{\min}).
\end{aligned} \tag{165}$$

$\square$

Note that when the matrix $Q = 0$ in Definition 5.1, all the steps in Lemma D.5 still hold, while $\varepsilon$ no longer matters since in (163) the right hand side is $\infty$, and the conclusion becomes

$$\mathbb{P}\left[\text{one or more irregular jumps in } [k - T_\varepsilon, k]\right] = 0 \tag{166}$$

which is also $O(T_\varepsilon \varepsilon)$.

**Lemma D.6.** *Under Assumption 5.6, the set $\Gamma(x)$ defined in (100) has $|\Gamma(x)| \geq 1, \forall x \in \mathcal{X} \setminus \{x_1\}$.*

*Proof.* We prove this by contradiction. Suppose there exists an $x \in \widetilde{\mathcal{X}} \setminus \{x_1\}$ such that $|\Gamma(x)| = 0$. According to the definition of $\Gamma(x)$ in (100), this implies that:

$$E\left(u_{\sigma^{(-n) \bmod M}(x)} - u_{\sigma^{(-n) \bmod M}(x_1)}\right) = 0, \forall n \in \{0, \cdots, M - 1\}. \tag{167}$$

Expanding this, we get:

$$\begin{aligned}
& E\left(u_x - u_{x_1}\right) = 0 \\
& E\left(u_{\sigma^{M-1}(x)} - u_{\sigma^{M-1}(x_1)}\right) = 0 \\
& E\left(u_{\sigma^{M-2}(x)} - u_{\sigma^{M-2}(x_1)}\right) = 0 \\
& \quad \vdots \\
& E\left(u_{\sigma(x)} - u_{\sigma(x_1)}\right) = 0.
\end{aligned} \tag{168}$$

This leads to:

$$\begin{bmatrix}
Eu_x - Eu_{x_1} \\
Eu_{\sigma(x)} - Eu_{\sigma(x_1)} \\
\vdots \\
Eu_{\sigma^{M-1}(x)} - Eu_{\sigma^{M-1}(x_1)}
\end{bmatrix} = 0. \tag{169}$$

This implies:

$$\widetilde{E}_s(u_x - u_{x_1}) = 0 \tag{170}$$

which means there are two columns in $\widetilde{E}_s$ that are identical. Thus, this contradicts Assumption 5.6b. $\square$

**Lemma D.7.** *Under Assumption 5.6, for each $n \in \Gamma(x)$, we have :*

$$d_n(x) \triangleq \mathbb{E}\left[g_{h_\varepsilon M - n}(x)\right] > 0. \tag{171}$$

*Proof.* We have that

$$
\begin{aligned}
&d_n(x) \\
&= \mathbb{E}\left[ \boldsymbol{g}_{h_\varepsilon M - n}(x) \right] \\
&= \mathbb{E}\left[ \log\left( \frac{\mathbb{P}(\boldsymbol{y}_{h_\varepsilon M - n} \mid \sigma^{(h_\varepsilon M - n) \bmod M}(x_1))}{\mathbb{P}(\boldsymbol{y}_{h_\varepsilon M - n} \mid \sigma^{(h_\varepsilon M - n) \bmod M}(x))} \right) \right] \\
&= \sum_{\boldsymbol{y}_{h_\varepsilon M - n} \in \mathcal{Y}} \mathbb{P}(\boldsymbol{y}_{h_\varepsilon M - n} \mid \sigma^{(h_\varepsilon M - n) \bmod M}(x_1)) \log\left( \frac{\mathbb{P}(\boldsymbol{y}_{h_\varepsilon M - n} \mid \sigma^{(h_\varepsilon M - n) \bmod M}(x_1))}{\mathbb{P}(\boldsymbol{y}_{h_\varepsilon M - n} \mid \sigma^{(h_\varepsilon M - n) \bmod M}(x))} \right) \\
&\overset{(a)}{>} 0,
\end{aligned}
\tag{172}
$$

where $(a)$ follows from the definition of $\Gamma(x)$ in (100), the two likelihood $\mathbb{P}(\boldsymbol{y}_{h_\varepsilon M - n} \mid \sigma^{(h_\varepsilon M - n) \bmod M}(x_1))$ and $\mathbb{P}(\boldsymbol{y}_{h_\varepsilon M - n} \mid \sigma^{(h_\varepsilon M - n) \bmod M}(x))$ have the same support, and since the two distributions are not the same, we always have $d_n(x) > 0$ based on the definition of Kullback–Leibler divergence. $\qquad\square$

**Lemma D.8.** *Under Assumption 5.6, let $\Lambda_n(t; x)$ be as defined in (99). It has the following properties for each $n \in \Gamma(x)$:*

*(1) $\Lambda_n(t; x) < \infty$ for all $t \in \mathbb{R}$ and is infinitely differentiable in $\mathbb{R}$.*

*(2) $\Lambda_n(0; x) = 0$, $\Lambda_n(-1; x) = 0$, and $\Lambda_n'(0; x) = d_n(x) > 0$.*

*(3) $\Lambda_n(t; x)$ is strictly convex in $\mathbb{R}$:*

$$
\Lambda_n''(t; x) > 0, \forall t \in \mathbb{R}.
\tag{173}
$$

*(4) $\Lambda_n(\tau; x) \geq d_n(x)\tau$ for all $t \in \mathbb{R}$.*

*(5) $\frac{\Lambda_n(t; x)}{t}$ is continuously differentiable in $\mathbb{R}$. In addition, we have:*

$$
\frac{d}{dt} \frac{\Lambda_n(t; x)}{t} > 0, \forall t \in \mathbb{R}.
\tag{174}
$$

*Proof.* We have:

(1) The statement follows from the definition of $\Lambda_n(t; x) < \infty$ and the fact that $\boldsymbol{g}_{h_\varepsilon M - n}(x)$ is a discrete random variable.

(2) The statement follows from simple calculations and Lemma D.7.

(3) The statement can be found in Matta et al. (2016, (115)).

(4) The statement can be proved by fist noticing that $\Lambda_n(\tau; x)$ is convex from (3) and the first-order convexity characterization:

$$
\Lambda_n(\tau; x) - \Lambda_n(0; x) \geq \Lambda_n'(0; x)(\tau - 0) \overset{(a)}{\Rightarrow} \Lambda_n(\tau; x) \geq d_n(x)\tau,
\tag{175}
$$

where $(a)$ follows from (2).

(5) The statement can be found in Matta et al. (2016, (119)).

$\qquad\square$

**Lemma D.9.** *Let $f(t)$ be twice continuously differentiable on $\mathbb{R}$, and define*

$$
g(t) \triangleq \frac{f(t)}{t}
\tag{176}
$$

*which is continuously differentiable on $\mathbb{R}$. Suppose further that $f(0) = 0$. Let $\tau_h \triangleq c(1-\mu)^h t$ where $0 < \mu < 1$ and $c > 0$. Then, for $t < 0$ it holds that*

$$\sum_{h=0}^{H-1} f(\tau_h) \leq \frac{1}{\mu}\left[\int_{c(1-\mu)^H t}^{ct} g(\tau)d\tau + \frac{\mu}{2-\mu}h_1(ct)\right], \tag{177}$$

*where*

$$h_1(t) \triangleq \max_{\tau \in [t,0]} |g'(\tau)|\frac{t^2}{2}. \tag{178}$$

□

*Proof.* Now we have $t < 0$. For $h \in \{0, \cdots, H\}$, we have

$$\tau_h = c(1-\mu)^h t \in [ct, c(1-\mu)^H t]. \tag{179}$$

Now let

$$G_h(t) \triangleq \int_{\tau_h}^{t} g(\tau)d\tau. \tag{180}$$

We can obtain:

$$G_h'(t) = g(t) \tag{181}$$
$$G_h''(t) = g'(t). \tag{182}$$

A second-order Taylor expansion gives:

$$\begin{aligned}
G_h(\tau_{h+1}) &\overset{(a)}{=} G_h(\tau_h) + G_h'(\tau_h)(\tau_{h+1} - \tau_h) + \frac{1}{2}G_h''(\bar{t}_h)(\tau_{h+1} - \tau_h)^2 \\
&\overset{(b)}{=} g(\tau_h)(\tau_{h+1} - \tau_h) + \frac{1}{2}g'(\bar{t}_h)(\tau_{h+1} - \tau_h)^2 \\
&\overset{(c)}{=} -g(\tau_h)\mu\tau_h + \frac{g'(\bar{t}_h)}{2}\mu^2\tau_h^2 \\
&\overset{(d)}{=} -f(\tau_h)\mu + \frac{g'(\bar{t}_h)}{2}\mu^2\tau_h^2,
\end{aligned} \tag{183}$$

where $(a)$ follows from the fact that $f(t)$ is twice continuously differentiable and Taylor's theorem with Lagrange remainder, here $\bar{t}_h \in [\tau_h, \tau_{h+1}]$; $(b)$ follows from the differentiation of $G_h(t)$ in (181) and (182); $(c)$ follows from $\tau_h$'s definition; $(d)$ follows from $g(t)$'s definition.

Then we have:

$$\begin{aligned}
\int_{ct}^{c(1-\mu)^H t} g(\tau)d\tau &= \sum_{h=0}^{H-1} G_h(\tau_{h+1}) \\
&= -\sum_{h=0}^{H-1} f(\tau_h)\mu + \sum_{h=0}^{H-1} \frac{g'(\bar{t}_h)}{2}\mu^2\tau_h^2.
\end{aligned} \tag{184}$$

Therefore,

$$
\begin{aligned}
\sum_{h=0}^{H-1} f(\tau_h) &= \frac{1}{\mu}\left[\int_{c(1-\mu)^H t}^{ct} g(\tau)d\tau + \sum_{h=0}^{H-1}\frac{g'(\bar{t}_h)}{2}\mu^2\tau_h^2\right] \\
&\stackrel{(a)}{\leq} \frac{1}{\mu}\left[\int_{c(1-\mu)^H t}^{ct} g(\tau)d\tau + \sum_{h=0}^{H-1}\left|\frac{g'(\bar{t}_h)}{2}\right|\mu^2 c^2(1-\mu)^{2h}t^2\right] \\
&\stackrel{(b)}{\leq} \frac{1}{\mu}\left[\int_{c(1-\mu)^H t}^{ct} g(\tau)d\tau + \mu^2 \max_{\tau\in[ct,0]}|g'(\tau)|\frac{(ct)^2}{2}\sum_{h=0}^{H-1}(1-\mu)^{2h}\right] \\
&\stackrel{(c)}{\leq} \frac{1}{\mu}\left[\int_{c(1-\mu)^H t}^{ct} g(\tau)d\tau + \frac{\mu}{2-\mu}\max_{\tau\in[ct,0]}|g'(\tau)|\frac{(ct)^2}{2}\right],
\end{aligned}
\tag{185}
$$

where $(a)$ follows from the definition of $\tau_h$; $(b)$ follows from the fact that $\bar{t}_h \in [\tau_h, \tau_{h+1}]$ and $\tau_h \in [ct, c(1-\mu)^H t] \subset [ct, 0]$; $(c)$ follows from the fact that $\sum_{h=0}^{\infty}(1-\mu)^{2h} = \frac{1}{2\mu-\mu^2}$. $\qquad\square$

**Lemma D.10.** *If $\delta_\varepsilon$ satisfies (31) and (32), and $\gamma_\varepsilon$ is defined as in (94), we have:*

$$
\lim_{\varepsilon\to 0}\gamma_\varepsilon = 0
\tag{186}
$$

$$
\lim_{\varepsilon\to 0}\frac{\delta_\varepsilon}{\gamma_\varepsilon} = \frac{1}{M}.
\tag{187}
$$

*Proof.* For the first one, we can easily prove by:

$$
\lim_{\varepsilon\to 0}\gamma_\varepsilon = \lim_{\varepsilon\to 0} 1 - (1-\delta_\varepsilon)^M = \lim_{\delta_\varepsilon\to 0} 1 - (1-\delta_\varepsilon)^M = 0.
\tag{188}
$$

For the second one, we have:

$$
\begin{aligned}
\lim_{\varepsilon\to 0}\frac{\delta_\varepsilon}{\gamma_\varepsilon} &= \lim_{\varepsilon\to 0}\frac{\delta_\varepsilon}{1-(1-\delta_\varepsilon)^M} \\
&= \lim_{\delta_\varepsilon\to 0}\frac{\delta_\varepsilon}{1-(1-\delta_\varepsilon)^M} \\
&\stackrel{(a)}{=} \lim_{\delta_\varepsilon\to 0}\frac{1}{M(1-\delta)^{M-1}} \\
&= \frac{1}{M},
\end{aligned}
\tag{189}
$$

where $(a)$ follows from L'Hôpital's Rule and the fact that $(\delta_\varepsilon)' = 1$ and $\left(1-(1-\delta_\varepsilon)^M\right)' = M(1-\delta_\varepsilon)^{M-1}$. $\qquad\square$

**Lemma D.11.** *Under Assumptions 5.3 and 5.6, we have $\|\widetilde{\boldsymbol{w}}_k\|_\infty \leq W$, where*

$$
\begin{aligned}
A_E &= \max_{y\in\mathcal{Y}}\|\log(\widetilde{E}^\top y)\|_\infty, \\
W &= \max\{\|\widetilde{w}_0\|_\infty, A_E\}.
\end{aligned}
\tag{190}
$$

*Proof.* From Assumption 5.6, we can see $A_E < \infty$ since $\widetilde{E} > 0$. From Assumption 5.3, we can get $W < \infty$. Given the dynamics in (24), we have:

$$
\begin{aligned}
\|\widetilde{\boldsymbol{w}}_k\|_\infty &= \|P(1-\delta)\widetilde{\boldsymbol{w}}_{k-1} + \delta\log(\widetilde{E}^\top\boldsymbol{y}_k)\|_\infty \\
&\leq \|P(1-\delta)\widetilde{\boldsymbol{w}}_{k-1}\|_\infty + \|\delta\log(\widetilde{E}^\top\boldsymbol{y}_k)\|_\infty \\
&= (1-\delta)\|P\widetilde{\boldsymbol{w}}_{k-1}\|_\infty + \delta\|\log(\widetilde{E}^\top\boldsymbol{y}_k)\|_\infty \\
&= (1-\delta)\|\widetilde{\boldsymbol{w}}_{k-1}\|_\infty + \delta\|\log(\widetilde{E}^\top\boldsymbol{y}_k)\|_\infty.
\end{aligned}
\tag{191}
$$

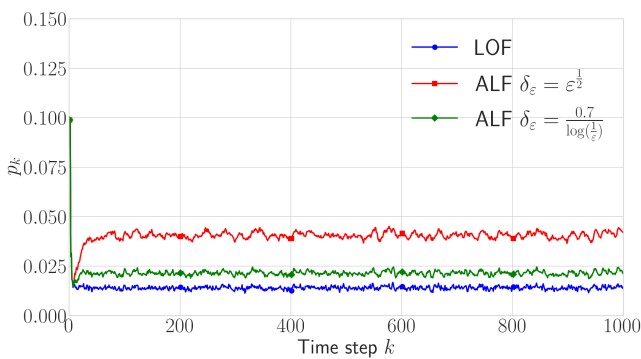

*Figure 7.* Per-time-step error probability for **ALF** and **LOF** under $\varepsilon = 0.005$.

Then we prove by induction. For $k = 0$, the statement is $\|\widetilde{w}_0\|_\infty \leq \max\{\|\widetilde{w}_0\|_\infty, A_E\}$, which is trivially true.

Assume the hypothesis is true for $k - 1$, i.e., $\|\widetilde{\boldsymbol{w}}_{k-1}\|_\infty \leq \max\{\|\widetilde{w}_0\|_\infty, A_E\}$. From (191), we have:

$$\begin{aligned}
\|\widetilde{\boldsymbol{w}}_k\|_\infty &\leq (1 - \delta)\|\widetilde{\boldsymbol{w}}_{k-1}\|_\infty + \delta A_E \\
&\leq (1 - \delta)\max\{\|\widetilde{w}_0\|_\infty, A_E\} + \delta \max\{\|\widetilde{w}_0\|_\infty, A_E\} \\
&= \max\{\|\widetilde{w}_0\|_\infty, A_E\}.
\end{aligned} \tag{192}$$

Thus, the hypothesis holds for $k$. By the principle of mathematical induction, it holds for all $k \geq 0$. □

## E. Additional Details for Section 6

### E.1. Illustrative Example

The state distribution is initialized as $\pi_0 = [1, 0]^\top$, and the initial logit vector as $w_0 = [0, 0]^\top$ for both **ALF** and **LOF**.

Let $\varepsilon = 0.005$. With 20,000 Monte Carlo iterations, the estimated instantaneous error probabilities for **LOF** and **ALF** are shown in Fig. 7. Since $w_0 = [0, 0]^\top$, the first update of **ALF** is given by $\boldsymbol{w}_1 = \delta_\varepsilon \log(E^\top \boldsymbol{y}_1)$, which implies that the decoding decision at $k = 1$ depends entirely on the observation. From the observation model in (47), the resulting error probability at the first timestep is 0.1. A similar argument holds for the **LOF** dynamics, which explains why all the curves in Fig. 7 start at approximately 0.1. We find that the estimated $p_k$ stablizes after 1000 steps, thus it's valid to use $p_{1000}$ to approximate $\limsup_{k \to \infty} p_k$.

It should be noted that the per-time-step error probabilities in Figure 7 and the long-run error behavior in Figure 1 exhibit similar patterns for more general deterministic matrices $D$ and when the underlying Markov chain involves more than two states. These additional results are carefully checked and omitted here for conciseness.

### E.2. RingWorld Game

#### E.2.1. SETTINGS

A sample transition matrix corresponding to CW1 is

$$T(\mathbf{CW1}) = P(\mathbf{CW1}) + \varepsilon Q(\mathbf{CW1}), \tag{193}$$

with entries for all $i, j \in \{1, \ldots, N\}$ defined as

$$p_{ij}(\mathbf{CW1}) = \begin{cases} 1, & i = (j \bmod N) + 1, \\ 0, & \text{otherwise,} \end{cases} \qquad q_{ij}(\mathbf{CW1}) = \begin{cases} -2, & i = (j \bmod N) + 1, \\ 1, & i \in \{(j - 1 \bmod N) + 1, (j + 1 \bmod N) + 1\}. \end{cases} \tag{194}$$

Let $\theta_j$ and $\xi_i$ denote the angles of state $j$ and beacon $i$, respectively. The observation model is

$$\mathbb{P}(\boldsymbol{y}_k = u_i \mid \boldsymbol{x}_k^\star = j) \propto \exp\big(\cos(\theta_j - \xi_i)\big). \tag{195}$$

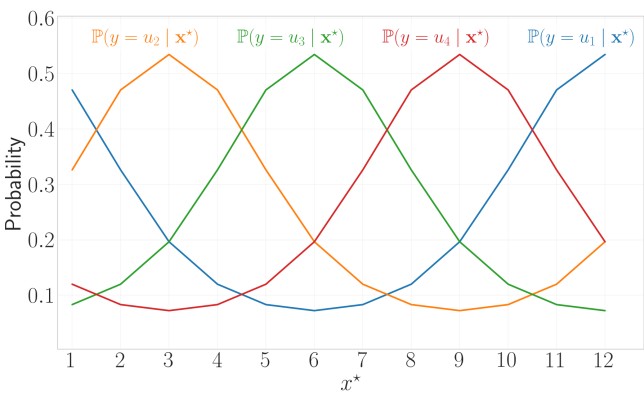

*Figure 8.* Observation probabilities $\mathbb{P}(y = u_i \mid \boldsymbol{x}^\star)$ for $i = 1, \ldots, 4$ versus $\boldsymbol{x}^\star \in \{1, \ldots, 12\}$, computed from (195).

and is further illustrated in Figure 8.

The reward function is defined as

$$r(\boldsymbol{x}_k^\star) = \frac{1}{K}\Big( \mathbb{1}\{\boldsymbol{x}_k^\star \text{ is a goal state}\} - \mathbb{1}\{\boldsymbol{x}_k^\star \text{ is a trap state}\} \Big), \tag{196}$$

where $K$ is the episode length. The total episodic reward therefore lies in $\sum_{k=1}^{K} r(\boldsymbol{x}_k^\star) \in [-1, 1]$.

Our PPO implementations, including the encoder, S5, actor, and critic, follow Lu et al. (2023). The general hyperparameters used for **LOF**, **ALF**, and **S5-based memory** in PPO training are summarized in Tables 1.

*Table 1.* Hyperparameters for training PPO

| Parameter | Value |
|---|---|
| Adam Learning Rate | 5e-5 |
| Number of Environments | 64 |
| Unroll Length | 1024 |
| Number of Timesteps | 30e6 |
| Number of Epochs | 30 |
| Number of Minibatches | 8 |
| Discount $\gamma$ | 0.99 |
| GAE $\lambda$ | 1.0 |
| Clipping Coefficient $\varepsilon$ | 0.2 |
| Entropy Coefficient | 0.0 |
| Value Function Weight | 1.0 |
| Maximum Gradient Norm | 0.5 |
| Learning Rate Annealing | None |
| Activation Function | LeakyReLU |
| Action Decoder Layer Sizes | [128, 128] |
| Value Decoder Layer Sizes | [128, 128] |

*Table 2.* Additional hyperparameters for S5-based memory

| Parameter | Value |
|---|---|
| S5 Discretization | ZOH |
| S5 $\Delta_{\min}$ | 0.001 |
| S5 $\Delta_{\max}$ | 0.1 |

For **S5-based memory**, the encoder layer sizes are $[128, 12]$ with recurrent hidden size 12, and $[128, 256]$ with hidden size 256. Additional fixed hyperparameters are shown in Table 2.

### E.2.2. ALF-INSPIRED DEEP LEARNING MEMORY

In our experiment, all permutation matrices $P(\text{CW1})$, $P(\text{CW2})$, $P(\text{CCW1})$, and $P(\text{CCW2})$ are circulant, meaning each row is a cyclic right shift of the previous one. According to Gray (2006, Theorem 3.1), any circulant matrix can be diagonalized by the discrete Fourier transform (DFT) matrix:

$$\begin{aligned} P(\text{CW1}) &= V\Lambda(\text{CW1})V^{-1}, \\ P(\text{CW2}) &= V\Lambda(\text{CW2})V^{-1}, \\ P(\text{CCW1}) &= V\Lambda(\text{CCW1})V^{-1}, \\ P(\text{CCW2}) &= V\Lambda(\text{CCW2})V^{-1}, \end{aligned} \tag{197}$$

where $V \in \mathbb{C}^{N \times N}$ is the DFT matrix, and each $\Lambda(\cdot) \in \mathbb{C}^{N \times N}$ is diagonal with eigenvalues lying on the complex unit circle. Specifically, $\Lambda(\text{CW1})$ has $N$ diagonal entries uniformly spaced on the unit circle. Moreover, since

$$P(\text{CW2}) = P(\text{CW1})^2, \quad P(\text{CCW1}) = P(\text{CW1})^\top, \quad P(\text{CCW2}) = (P(\text{CW1})^\top)^2, \tag{198}$$

we obtain

$$\begin{aligned} \Lambda(\text{CW2}) &= \Lambda(\text{CW1})^2, \\ \Lambda(\text{CCW1}) &= \Lambda(\text{CW1})^{-1}, \\ \Lambda(\text{CCW2}) &= \Lambda(\text{CW1})^{-2}. \end{aligned} \tag{199}$$

Then, ALF in (46) can be rewritten as

$$\begin{aligned} V^{-1}\boldsymbol{w}_k &= \Lambda(\boldsymbol{a}_{k-1})(1-\delta)\,(V^{-1}\boldsymbol{w}_{k-1}) + \delta V^{-1}\log(E^\top \boldsymbol{y}_k), \\ \underbrace{\boldsymbol{w}_k}_{\text{to actor and critic}} &= V(V^{-1}\boldsymbol{w}_k). \end{aligned} \tag{200}$$

Let

$$\boldsymbol{h}_k \triangleq V^{-1}\boldsymbol{w}_k, \tag{201}$$

so that

$$\begin{aligned} \boldsymbol{h}_k &= \Lambda(\boldsymbol{a}_{k-1})(1-\delta)\,\boldsymbol{h}_{k-1} + \delta V^{-1}\log(E^\top \boldsymbol{y}_k), \\ \boldsymbol{w}_k &= V\boldsymbol{h}_k. \end{aligned} \tag{202}$$

This transformation maps the real-valued state dynamics into the complex diagonal space, which is how linear RNNs are usually implemented. In this complex space, we aim to develop ALF-inspired deep learning memory architectures by treating $\Lambda(\cdot)$, $\delta$, $E$, and $V$ as learnable parameters. We refer to these architectures collectively as **Deep ALF**.

We first make all parameters learnable, initializing $\Lambda(\cdot)$, $E$, and $V$ with their true values, and setting $\delta = 0.1$ to the empirically good value used in Section 6.2. The results are compared with the original **ALF** (with $\delta = 0.1$) and the softmax-processed **LOF** in Figure 9.

From Figure 9, we observe that making all parameters learnable allows **Deep ALF** to slightly outperform **ALF**, reaching performance levels close to the softmax-processed **LOF**.

Next, we keep $\Lambda(\cdot)$, $E$, $V$, and $\delta$ learnable, but vary their initialization schemes:

- **Random $E$:** $\Lambda(\cdot)$ and $V$ are initialized with their true values, and $\delta = 0.1$. For $E$, we first draw $\kappa \in \mathbb{R}^{S \times N}$ from a standard normal distribution and then normalize it column-wise:

$$E = \text{softmax}(\kappa) \quad \text{(column-wise)}. \tag{203}$$

This setup represents the case where the observation model is unknown, but other environment parameters are known.

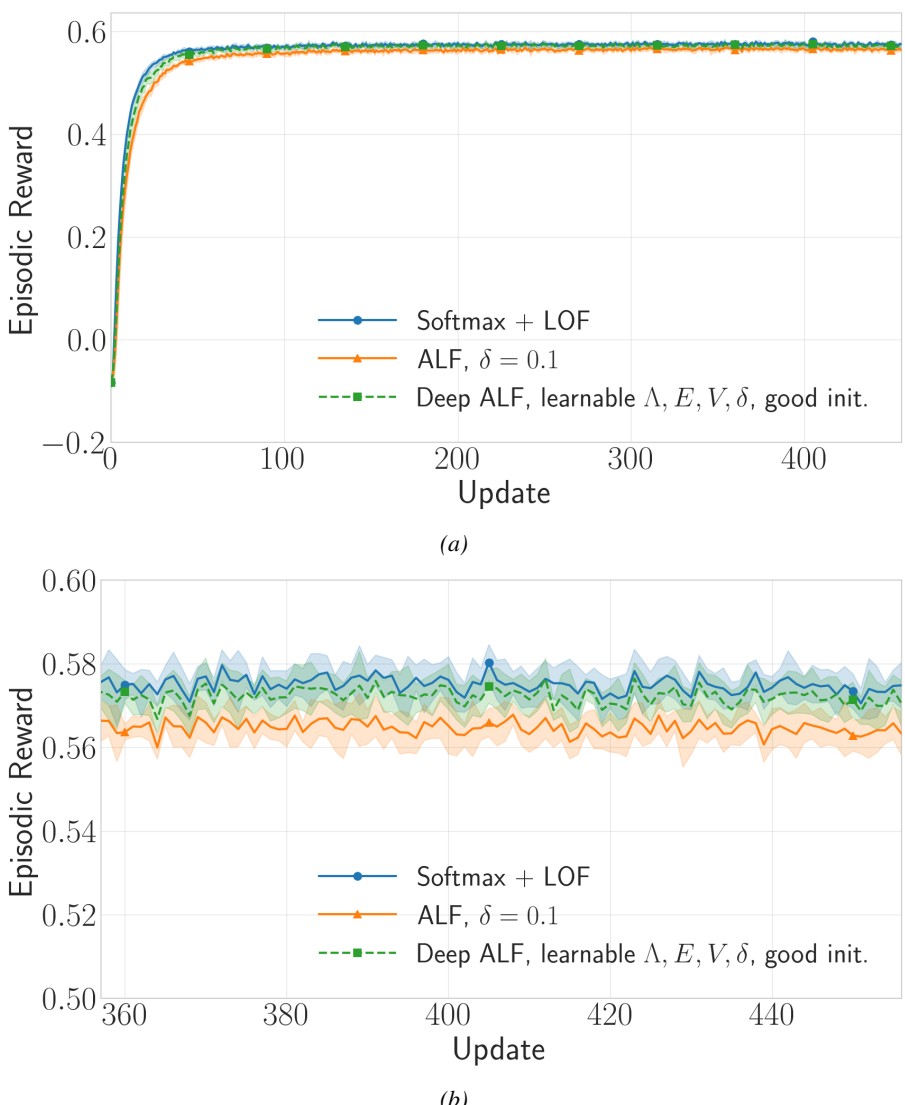

*Figure 9.* Episodic return versus update. (a) shows the full results, and (b) shows a zoomed-in view of the last 100 updates. The shaded area indicates the standard deviation across 8 random seeds.

- **Random $\delta$:** $\Lambda(\cdot)$, $E$, and $V$ are initialized with their true values, while $\delta$ is drawn uniformly from $(0, 0.5)$. This corresponds to an unknown perturbation pattern $\varepsilon Q$, since the optimal $\delta$ should depend on the specific $\varepsilon Q$.

- **Random $\Lambda(\cdot)$:** $E$ and $V$ are initialized with their true values, and $\delta = 0.1$. Each eigenvalue of $\Lambda(\cdot)$ is initialized on the complex unit circle with uniformly random phases, representing an unknown permutation transition pattern.

- **All random:** $\Lambda(\cdot)$, $E$, and $\delta$ follow the random initialization schemes above. For $V$, each element is initialized with independent standard normal real and imaginary parts. This setting models a fully unknown environment where only the numbers of states $N$ and observations $S$ are known.

From Figure 10, we observe that random initialization of $E$ or $\delta$ still enables effective learning, achieving performance close to that with full informed initialization. Random $\Lambda(\cdot)$ causes a slight degradation in performance.

In addition, when all parameters $\Lambda(\cdot)$, $E$, $V$, and $\delta$ are randomly initialized, making this setting directly comparable to **S5-based memory**, Figure 10a shows that the parameterization of **Deep ALF** is more efficient and yields better performance than **S5-based memory**: With the same hidden size of 12, both **Deep ALF** and **S5** converge quickly, but **Deep ALF**

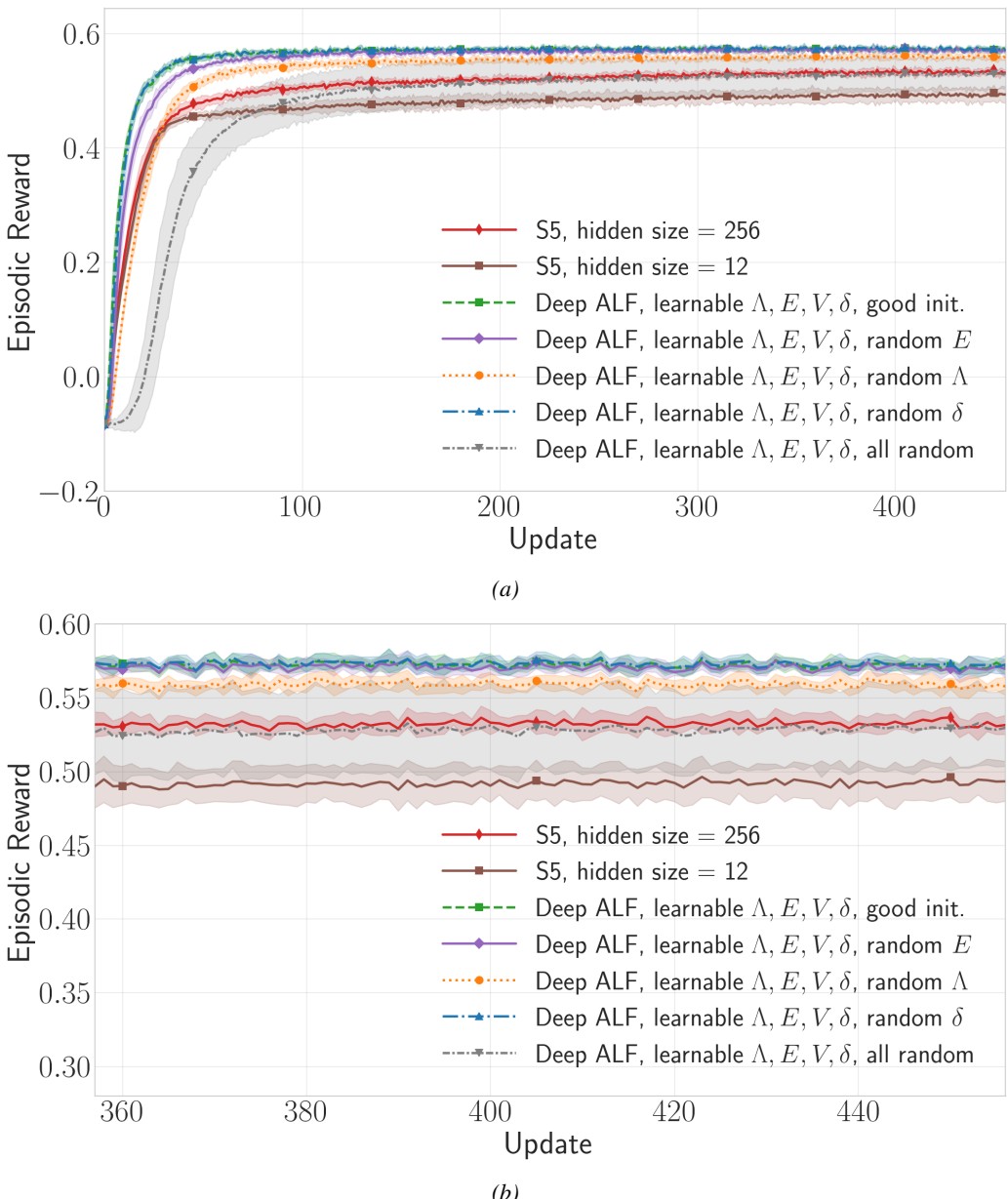

*(a)*

*(b)*

*Figure 10.* Episodic return versus update. (a) shows the full results, and (b) shows a zoomed-in view of the last 100 updates. The shaded area indicates the standard deviation across 8 random seeds.

achieves noticeably better final performance, as shown in Figure 10a. Only by increasing **S5**'s hidden size to 256 (with substantially more parameters) does its performance approach that of **Deep ALF**. Furthermore, **Deep ALF** offers a far more efficient encoder parameterization. The **S5** agent uses a two-layer MLP encoder with $2048\,(4 \times 128 + 128 \times 12)$ parameters for hidden size 12 and $33{,}280\,(4 \times 128 + 128 \times 256)$ parameters for hidden size 256. In contrast, **Deep ALF** uses the encoder $\log(E^\top y)$ (Equation (202), with $E$ learned from scratch, with only $48\,(4 \times 12)$ parameters, due to its well-designed dynamics. These results demonstrate that **Deep ALF** achieves better performance with significantly fewer parameters.

However, note that our transformation from **ALF** in (46) to **Deep ALF** in (202) relies on all permutation matrices in this experiment being circulant, such that a common DFT matrix $V$ diagonalizes them. In general, while permutation matrices are diagonalizable, they do not necessarily share the same diagonalization matrix $V$. Designing ALF-inspired deep learning memory architectures for more general nearly-permutation environments is an interesting direction for future work.

