# OpenReview forum: "Why Linear Recurrent Memory Works in Partially Observable Reinforcement Learning"
_ICML.cc/2026/Conference — ICML 2026 spotlight_

### Official Review · Reviewer_fvcr · 2026-03-01

**Soundness:** 4
**Presentation:** 4
**Significance:** 4
**Originality:** 3
**Overall Recommendation:** 5
**Confidence:** 3

**Summary:**

The paper asks a focused question: why do linear recurrent memories, such as linear RNNs and state space model style memories, work surprisingly well for partially observable reinforcement learning. The authors study this through logit space filtering for Hidden Markov models and action controlled Hidden Markov models. They give two constructive results. First, for deterministic transitions, including permutation dynamics and a broader deterministic class with transient states, they show that a linear recurrence can exactly reproduce the optimal pre softmax belief logits. In this setting, the linear recurrent state is effectively a sufficient statistic for optimal control because it matches the Bayesian log belief update. Second, for nearly deterministic transitions of the form $T = D + \epsilon Q$ they propose an Adaptive Logit Filter that blends a discounted, permutation aligned memory update with current observation log likelihoods. Under mild multi step observability conditions and an appropriate adaptation schedule for the mixing parameter, they prove the long run state decoding error vanishes with an $\epsilon \log(1/\epsilon)$ rate that matches known optimal asymptotics for belief based decoding. They extend the construction to the action controlled case by making the linear dynamics depend on the action.

The paper includes a small synthetic Hidden Markov model experiment to illustrate the asymptotic behavior of the filter under different schedules, and a RingWorld partially observable control task to show the filter can be used as a stable feature or target for policy optimization.

**Compliance With Llm Reviewing Policy:**

Affirmed.

**Final Justification:**

Thank you for the detailed rebuttal. My main concerns have been addressed, and I therefore maintain my original overall assessment.

**Key Questions For Authors:**

See weaknesses.

**Limitations:**

yes

**Strengths And Weaknesses:**

### Strengths

* The paper gives a clean exactness result for deterministic transitions, showing that the optimal log belief logits can be reproduced by a linear recurrence when the transition is a permutation, and extends this to deterministic matrices with transient states via a finite horizon correction.
* For nearly deterministic transitions of the form $T = D + \epsilon Q$, the proposed Adaptive Logit Filter has a clear update rule that balances retained memory and new evidence, and the analysis provides a vanishing decoding error guarantee with an $\epsilon \log(1/\epsilon)$ rate under observability and schedule conditions.
* The logit space viewpoint makes the gap between Bayesian filtering and linear recurrence very explicit.
* The work connects adaptive social learning style ideas to modern linear recurrent memories in RL in a concrete, constructive way, and packages them into a filter that is easy to reason about and directly usable as a feature in policy learning.

### Weaknesses

* The deterministic exactness relies on a time indexed input map psi that depends on the step index and a short observation window, which reads more like a proof level construction than what a standard learned encoder would implement.
* The ALF guarantees assume access to the emission model structure and the permutation aligned dynamics, and the transient state analysis uses a minus infinity style initialization for transient coordinates, so the path from theory to realistic learning settings is not fully addressed.
* The scope is concentrated on deterministic or nearly permutation dynamics; the paper is upfront that extending to more general POMDPs with substantial stochasticity or more complex action dependent transients is nontrivial, so the conclusions should be interpreted as strongest in that structured regime.
* The empirical evaluation is illustrative but narrow, and the comparisons mix a hand specified filter that uses environment knowledge with learned memories, with limited coverage of strong nonlinear recurrent baselines like GRU or LSTM under matched capacity and training budgets.

---

> ### Author Rebuttal · Authors · 2026-03-31
>
> Thank you very much for your positive review! Below, we respond to your critical points.
>
> **Deterministic exactness looks like a proof level construction**
>
> We agree that the structure of $\psi$ in our setting is uncommon and worth further investigation. However, in
> partially observable reinforcement learning, using a fixed window of past observations as input to a memory unit
> is a common and effective approach (Eberhard et al., 2025; Efroni et al., 2022). It helps mitigate the limited
> observability of single observations and, under suitable assumptions, can even enable exact recovery of the
> underlying states. To address the reviewer’s concern, we will add the following statement after Theorem 4.4:
> ''Although $\psi$ is a nonlinear mapping, its structure is uncommon in deep learning and is not differentiable with respect to $k$. Understanding how well standard architectures, such as MLPs, can approximate this mapping is an interesting direction for future work, especially since nonlinear encoders are typically implemented using MLPs.''
>
>
> **From theory to realistic learning**
>
> With known environment parameters, the constructed **ALF** serves as an effective proxy for policy learning. In addition, we also implemented an ALF-inspired deep learning memory, **Deep ALF**, in Appendix D.2.2, which preserves the ALF structure while treating all parameters as learnable. Even without access to environment parameters, **Deep ALF** still achieves better performance than the **S5-based memory** in this experiment, while using significantly fewer parameters.
>
> The $-\infty$ initialization for transient coordinates is introduced for analytical convenience. In practice, it can be replaced by a sufficiently large negative value. Since $\delta$ is typically small and $1-\delta$ is close to 1, the values of these transient coordinates remain suppressed throughout the episode and are unlikely to affect decoding. Therefore, this assumption can be handled without difficulty in practical implementations.
>
> For a discussion on relaxing Assumption 5.6(a), please refer to the section “Relaxing Assumption 5.6(a)” in our response to Reviewer 2VqL.
>
> **Limited scope and extension to highly-stochastic environments**
>
> Please see the section "Restrictive determinism assumption and limited evaluations" in our response to Reviewer neRB.
>
> **Comparisons with GRU or LSTM**
>
> As noted in our previous response, linear RNNs have been shown to outperform GRU, LSTM, and Transformers in multiple partially observable benchmarks. The focus of this manuscript is to understand why linear recurrent memory is effective, rather than to propose new memory architectures. In this context, introducing additional memory units such as Transformers or nonlinear RNNs would not directly contribute to the main objective and may dilute the focus of the paper. That said, we agree that a comprehensive empirical comparison between Deep ALF and other widely used memory architectures would still be valuable. We consider this a meaningful direction for future work.

---

> > ### Author Rebuttal · Reviewer_fvcr · 2026-04-01
> >
> > Thank you for the detailed rebuttal. My main concern has been sufficiently addressed, and I will maintain my decision for this paper.

---

> > > ### Author Response · Authors · 2026-04-03
> > >
> > > Thank you very much for acknowledging our paper and response. We really appreciate your time and helpful comments!

---

### Official Review · Reviewer_EPMz · 2026-03-11

**Soundness:** 3
**Presentation:** 3
**Significance:** 3
**Originality:** 3
**Overall Recommendation:** 4
**Confidence:** 5

**Summary:**

This paper shows that linear-memory recurrent neural networks (RNNs) or state-space models (SSMs) can improve reinforcement learning in partially observable environments because they can infer the current state from historical memory from a theoretical perspective, thereby compensating for important information missing from local observations.

**Compliance With Llm Reviewing Policy:**

Affirmed.

**Final Justification:**

I appreciate the new evidence the authors provided during rebuttal phase. I have raised my score to 4.

**Key Questions For Authors:**

1. If the authors add visualizations (or other qualitative evidence) showing that the model indeed captures the underlying state, I will raise my score by 1 point.
2. If the authors include experiments on mainstream benchmarks such as Atari, I will raise my score by 1 point.

**Limitations:**

Yes.

**Strengths And Weaknesses:**

Strengths:
1. The theoretical derivations are rigorous, and the conclusions are credible.
2. The experimental results are consistent with and support the paper’s claims.
3. While the proposed perspective has limitations (e.g., it may not apply well to highly stochastic environments), these limitations are reasonable: in strongly stochastic settings, modeling history with sequence models is unlikely to yield strong gains anyway. Importantly, the paper does not shy away from this issue and instead delineates and discusses the scope clearly.

Weaknesses:
1. The empirical evidence is somewhat limited and lacks qualitative analyses. In particular, the paper would benefit from visualizations demonstrating that the linear-memory network indeed captures (or reconstructs) the underlying current state.
2. The evaluation scope is relatively narrow. Extending the experiments to mainstream benchmarks such as Atari would provide stronger support for the paper’s claims. Atari is pixel-based and largely deterministic, which seems especially well aligned with the paper’s setting and assumptions.

---

> ### Author Rebuttal · Authors · 2026-03-31
>
> Thank you very much for your review! Below, we respond to your critical points and questions.
>
> **Showing linear memories recover the true states**
>
> We agree with the reviewer that explicitly showing how linear models recover the underlying states would improve clarity and make our results more accessible. To this end, we will add a new appendix section titled ''Visualization of ALF's state-tracking'':
>
> - Time-invariant ALF:
>
> In this section, we consider a two-state HMM:
> $$
> T = [1-\varepsilon, \varepsilon; \varepsilon, 1-\varepsilon], \quad E = [0.8, 0.2; 0.2, 0.8]
> $$
> with $\varepsilon = 0.005$. In this setting, the sign of the difference $\boldsymbol{w}_k(x_2)-\boldsymbol{w}_k(x_1)$ fully determines the decoded state (positive implies $x_2$, and vice versa). We therefore focus on the evolution of this difference.
>
> Two observations can be drawn from Figure (https://anonymous.4open.science/r/ICML2026-Linear-Memory-753F/HMM-states.png). First, although the overall trends are similar, there is a clear discrepancy between the dynamics of **ALF** and **LOF**. Nevertheless, **ALF** still achieves good state decoding. Second, when $\varepsilon = 0.005$, we have $\varepsilon^{1/2} < 1/\log(1/\varepsilon)$. As a result, the dynamics under $\delta_\varepsilon = \varepsilon^{1/2}$ appear smoother than those under $\delta_\varepsilon = 1/\log(1/\varepsilon)$, since a smaller $\delta_\varepsilon$ reduces sensitivity to instantaneous (and noisy) observations. However, this smoother behavior comes at the cost of slower adaptation to irregular state transitions, as reflected by the lag after switching points (e.g., $k = 115, 270, 406, 593$). This reflects the intrinsic tradeoff between stability and responsiveness in ALF.
>
> - Time-varying ALF:
>
> We next consider the RingWorld environment (Section 6.2). Figure (https://anonymous.4open.science/r/ICML2026-Linear-Memory-753F/RingWorld-states.pdf) shows a sample trajectory of true states, estimated states, and observations. The **ALF** is constructed using environment knowledge with $\delta = 0.1$, and the corresponding trained actor is used to select actions. In the legend, *Irregular Transition* refers to state transitions that deviate from the permutation backbone; *Off-peak Observation* indicates observations that do not match the most likely ones given the current state; and *Decoding Error* denotes mismatches between the decoded and true states.
>
> Two observations follow. First, **ALF** is more sensitive to noisy observations than **LOF**. For example, at timesteps $61$ and $69$, observations $B_3$ and $B_4$ occur at state 3, which are unlikely; these lead to decoding errors for **ALF** but not for **LOF**. Second, the learned policy is effective: the agent takes $CW1$ at state 2 and $CCW1$ at state 3 for most of the time, which leads to high return.
>
> **Evaluation on mainstream benchmarks**
>
> Indeed, the environments we consider in our experiments are more illustrative than realistic. The main contributions of this manuscript are fundamental, and we are working on a follow-up project to test our conclusions in more complex environments. As noted by the reviewer, deterministic or nearly-deterministic dynamics are actually common in the existing benchmarks. Even some continuous environments (e.g., Mountain Car in POPGym or robotic control tasks in MuJoCo) can be approximated by finite, nearly deterministic POMDPs through discretization of the state, action, and observation spaces. However, these evaluations would require careful engineering and additional considerations (e.g., discretization accuracy). A proper treatment of these issues, therefore, goes beyond the scope of this manuscript.

---

> > ### Author Rebuttal · Reviewer_EPMz · 2026-04-01
> >
> > Thank the authors for the rebuttal. I appreciate the authors’ response and the additional qualitative evidence regarding state tracking.
> >
> > In particular, the added visualizations and accompanying discussion are helpful for addressing my first concern. They provide more concrete evidence that the proposed linear-memory model is not only improving return, but is also tracking or decoding the underlying state in a meaningful way. This strengthens the empirical support for the paper’s central mechanism and addresses the qualitative-analysis gap I pointed out in my original review.
> >
> > At the same time, my concern about the limited experimental scope on more mainstream benchmarks remains. The rebuttal explains why extending the evaluation to such settings would require additional engineering and modeling choices, but this aspect is still not resolved in the current submission.
> >
> > Overall, the rebuttal meaningfully improves my assessment by addressing the requested qualitative evidence on state recovery. I am therefore increasing my score by 1 point.

---

> > > ### Author Response · Authors · 2026-04-03
> > >
> > > Evaluating our approach on more realistic environments is an important direction, and we are currently working on it. Thank you again for your time in reviewing and responding!

---

### Official Review · Reviewer_neRB · 2026-03-12

**Soundness:** 3
**Presentation:** 3
**Significance:** 3
**Originality:** 2
**Overall Recommendation:** 5
**Confidence:** 3

**Summary:**

The paper investigates why linear recurrent models seem to work effectively in POMDPs even though the belief state is nonlinear. A key idea is that when the transition matrix is a deterministic matrix, the non-linearity in the unnormalized belief vanishes. The paper then proposes adaptive logit filter for nearly-deterministic transitions which has the form of a linear recurrence, w_k = P(1-δ)w_{k-1} + δ log(E⊤y_k), and shows that when  δ is chosen properly, the long term error vanishes at rate ε·log(1/ε).

**Compliance With Llm Reviewing Policy:**

Affirmed.

**Final Justification:**

The authors' rebuttal had addressed all my concerns.

**Key Questions For Authors:**

- The results show that linear recurrent memory suffices when the transition matrix is (near) deterministic. However, previous work [1] showed that the representation equivalence between dense and linear diagonal RNNs does not extend to the non-linear case, suggesting that things break when the near determinism assumption doesn’t hold. Can you discuss more precisely where the boundary lies between environments where linear memory is provably sufficient and those where nonlinear recurrence becomes necessary?
- Have you observed ALF-like structure (permutation-like dynamics matrices, (1-δ)/δ balance) in linear RNNs trained end-to-end on POMDPs?
- How does your analysis relate to the state-tracking results showing linear RNNs need complex eigenvalues for permutation tracking? I see some relations to the unit circle eigenvalues and i am wondering if you can elaborate on that.

[1] Elelimy, E., White, A., Bowling, M., & White, M. (2024). Real-time recurrent learning using trace units in reinforcement learning. Advances in Neural Information Processing Systems.

**Limitations:**

the paper didn't discuss the limitations (except for acknowledging the simple environments used for the experiments). Please check the weaknesses section for the limitations I noticed.

**Strengths And Weaknesses:**

### Strengths:
- The paper identifies precise conditions (deterministic/near-deterministic transition) under which linear RNNs become theoretically justified for POMDPs.
- The main result (theorem 5.7) shows that the error has an optimal rate ε·log(1/ε) that matches optimal nonlinear filters.
- The insights around δ/(1-δ) balance, and the analogy between linear RNNs and ALF with eigenvalues near the unit circle are helpful.

### Weaknesses:
- The determinism assumption is very restrictive; many real-world POMDPs have stochastic transitions.
- The constructed filter requires knowledge of the environment which is impractical.
- The experimental analysis is limited; only two-state mdps and ring world game. There are many benchmarks for POMDPs (that are not too computationally intensive) that would have strengthened the results such as POPGYM [1].

[1] Steven Morad, Ryan Kortvelesy, Matteo Bettini, Stephan Liwicki, and Amanda Prorok. POPGym: Benchmarking partially observable reinforcement learning. In International Conference on Learning Representations, 2023.

---

> ### Author Rebuttal · Authors · 2026-03-31
>
> Thank you very much for your positive review! Below, we respond to your critical points.
>
> **Restrictive determinism assumption and limited evaluations**
>
> Our discussions focus on (near-)deterministic dynamics, however they are not uncommon in existing benchmarks, and linear recurrent networks have been shown to perform well in such settings. For example, the linear recurrent unit outperforms LSTM on the Passive T-Maze task in Lu et al. (2024). Similarly, S5 achieves the best return on the deterministic POPGym task ''RepeatPreviousHard'' (see Table 5 in Lu et al. (2023)). In addition, although our analysis focuses on settings with finite state, observation, and action spaces, many continuous control problems can be approximated by finite, nearly deterministic POMDPs through discretization (e.g., robotic control tasks in MuJoCo). We are currently working on extending our evaluation to these more complex environments. However, such experiments require careful engineering and are beyond the scope of this manuscript.
>
> Also, we note that under highly stochastic environments, the state-deconding performance of sequence models is not likely to yield strong gain anyways (also noted by Reviewer EPMz). For example, in the fully stochastic case with $\varepsilon = 0.5$ in Section 6.1, the first term of **LOF** update (7) becomes proportional to the all-one vector. In this case, the optimal filter reduces to a myopic estimator, i.e., decoding based solely on $\log(E^\top y)$, and the past history no longer provides useful information. Therefore, in highly stochastic environments, it is unclear whether it's beneficial to use linear RNNs or other sequence models.
>
> **Environment knowledge in the constucted filter**
>
> It is true that the linear filters evaluated in Section 6 are fixed and constructed from environment knowledge. However, as noted in the last paragraph of Section 6.2, we also implement an ALF-inspired deep learning model, **Deep ALF**. When the environment parameters are unknown, **Deep ALF** still achieves better performance than the **S5-based memory**, while using significantly fewer parameters. Moreover, when the environment parameters are partially known, **Deep ALF** can leverage this information to obtain improved initialization (see Figure 8). In contrast, it remains unclear how such prior knowledge can be incorporated into existing linear recurrent networks.
>
> **The boundary of near-derterminstic assumption and linear memories**
>
> We agree that clarifying this boundary is important. Our results should be interpreted as identifying a sufficient regime for linear memory, rather than a universal characterization of all partially observable environments. In particular, linear memory is provably effective when the transition dynamics are close to a deterministic backbone structure. In this regime, the state evolution remains sufficiently close to the backbone and can therefore be captured by a linear update. When this regime breaks, the state transition terms no longer provide useful information and a nonlinear LOF-like memory might be needed.
>
> Finally, we note that the notion of ''near-deterministic'' depends not only on $\varepsilon$ but also on the structure of $Q$. For instance, in Experiment 6.1, $\varepsilon$ can be as large as $0.1$, if $Q$ is smaller, $\varepsilon$ can be larger accordingly.
>
> **ALF-like structure in linear RNNs**
>
> The $(1-\delta)$/$\delta$ balance is commonly observed in linear RNNs trained on POMDPs. For example, the Linear Recurrent Unit (LRU) adopts a similar structure to mitigate forward-pass instability (see Figure 1 in Orvieto et al., (2023)). Moreover, LRU has been shown to outperform GPT and LSTM on multiple benchmarks and is described as “a well-suited alternative for partially observable RL” (Lu et al., 2024).
>
> Regarding permutation-like dynamics, we note that although existing linear RNNs are designed to model real dynamical systems, many operate in the complex domain for computational efficiency. As a result, it is often difficult to identify a clear dynamical system counterpart. Consequently, it is unclear whether such models implicitly implement permutation-like dynamics.
>
> **Deep ALF operates in complex domain**
>
> In Appendix D.2.2, since permutation matrices are diagonalizable, we follow a standard approach to transform the real-valued **ALF** into its complex-domain equivalent (196), thereby making all **ALF** parameters learnable and yielding **Deep ALF**. The resulting **Deep ALF** builds on our analysis and state-tracking guarantees for **ALF**, and achieves improved performance with greater parameter efficiency compared to the **S5-based memory**.
>
> Moreover, since all permutation matrices have eigenvalues with unit magnitude, the state transition matrix $(1-\delta)P(a)$ of **ALF** in permutation-backboned environments has eigenvalues whose magnitudes are close to one (scaled by $1-\delta$), i.e., they lie near the complex unit circle.

---

> > ### Author Rebuttal · Reviewer_neRB · 2026-04-03
> >
> > Thank you for the thoughtful rebuttal, that cleared my understanding. I will increase the score to 5.

---

> > > ### Author Response · Authors · 2026-04-03
> > >
> > > Thank you very much for updating the evaluation. We really appreciate your time and helpful comments!

---

### Official Review · Reviewer_2VqL · 2026-03-13

**Soundness:** 3
**Presentation:** 3
**Significance:** 2
**Originality:** 2
**Overall Recommendation:** 5
**Confidence:** 2

**Summary:**

This paper studies the representational ability of linear recurrent neural networks for implementing bayes optimal policies.
By restricting the class of POMDP to POMDPs with nearly-deterministic transitions, the paper shows that the belief filter can be (approximately) implemented by certain linear filters (corresponding to possible implementations of linear recurrent neural networks).
More precisely, it is shown for deterministic transition functions that a linear filter can exactly implement the optimal belief filter (or rather, the logits of this optimal belief, which is linear for this particular instance of POMDP).
Then, the paper proposes another filter adapted to POMDP with nearly-deterministic transition functions, for which the filter is not optimal but present some error.
It is shown that the asymptotic error (for sufficiently large history) of this belief filter is bounded by a term that increases with epsilon, where epsilon characterizes the transition noise.
The paper also bounds the rate of convergence of the error with epsilon.
These results also come with assumptions on the emission process.
Finally, it is empirically demonstrated that this filter provide a better learning for the RL algorithm, compared to other (learnable) recurrent modules, but it requires the transition and emission function to be known.
They also propose a learnable version of this filter that provides better convergence for the algorithm compared to S5.

**Compliance With Llm Reviewing Policy:**

Affirmed.

**Final Justification:**

The rebuttals and discussions with the other reviewers confirmed my original assessment, which is acceptance.

**Key Questions For Authors:**

- In the second paragraph of section 5, when you say that $w_k$ is not necessarily a sufficient statistic, you do refer to the nearly-deterministic setting, isn't it? Because the $w_k$ used in the nearly-deterministic setting was not introduced yet, which makes it confusing. And what do you mean by "consistently reproducing the true state"? The argmax predictor will still make mistakes in this nealry-deterministic setting, isn't it?
- Why does (14) depends on the full history of past observations and (18) only depends on the last action and observation? If I understood correctly, it is because this second result is given in the case where $r = 0$, while the first one is given in the general case. If yes, it should maybe be stated more clearly.
- How difficult would it be to relax assumption 5.6 (a)? Relaxing this assumption (of having full support for the observation distribution in any state) seems to make the inference process easier. I was thus wondering whether this was just to avoid technicalities in the proof, and whether it could easily be relaxed?

**Strengths And Weaknesses:**

- *Soundness*. The paper is supporting its claims and the theoretical results seem correct, the empirical demonstration is convincing.
    - I note that the results of Figure 3 are comparing policies that use (near-)optimal filters as input to baselines that learn these filters without additional knowledge. It is a bit unfair, and it should be better acknowledged that your methods have more environment knowledge and that it could partially explain why it works so well.
    - Could you add the return of *Deep ALF* in Figure 3? In my opinion, this would be the interesting version to compare to S5 (because both methods learn the filter in that case).
- *Presentation*. The paper is clear and well presented. The results are very accessible, despite the technicality of the proofs. Indeed, while many details are available in appendices, the main text is mostly self-contained.
    - I would have liked to read a presentation and discussion of the $\xi$ constant and read an explanation of how you verify that (48) verifies (34).
- *Significance*. I have a concern regarding the significance of the result. Is the class of nearly-deterministic POMDP large enough? Does it really encompasses the environments for which the SSM backbone has obtained good results in previous works? That said, I still think that the analysis is worthwhile and give a good idea of the representations that might be learned by such backbones when used in general POMDPs, along with an idea of their associated errors.
- *Originality*. This paper is targeting a novel analysis in a new setting, even though adapting several results from Khammassi et al. (2025).

---

> ### Author Rebuttal · Authors · 2026-03-31
>
> Thank you very much for your positive review! Below, we respond to your critical points.
>
> **Clarification on environment knowledge in ALF/LOF and inclusion of Deep ALF**
>
> In Fig. 3, **ALF/LOF** are constructed using known environment knowledge, whereas **S5** is trained from scratch. We agree that this distinction should be stated clearly to avoid potential confusion. We also think it's a good idea to include Deep ALF (i.e., the all random one in Fig. 8) in Fig. 3. Accordingly, we will revise the paragraph starting from Line 426 as follows: ''... four types of memory: **LOF**, **ALF**, an ALF-inspired deep learning variant called **Deep ALF**, and an **S5-based memory** ... For both **S5-based memory** and **Deep ALF**, the memory parameters are randomly initialized and jointly trained with the actor and critic.'' Finally, we will revise the caption of Fig. 3 by adding: ''**ALF/LOF** are constructed from environment knowledge, whereas **S5** and **Deep ALF** are trained from scratch.''
>
> **Clarification and discussion of the constant** $\xi$
>
> The constant $\xi$ plays an important role in this state-decoding problem, and we agree that it should be discussed in more details. Therefore, we will revise the sentence following (34) as ''where the constant $\xi>0$ depends only on the permutation backbone $P$ (see (22)) and the likelihood model $E$. A detailed definition of $\xi$, along with further discussion, is provided in Appendix C.1. Then it holds that ...''. We will also revise the sentence following (48) as ''The latter choice also satisfies (34); the detailed derivations are provided in Appendix C.1.''
>
> In Appendix C.1, we will add a short discussion on $\xi$: ''As shown in (138), $\xi$ depends on $M$, the order of the permutation matrix $P$, and $\Lambda_n$, the moment generating function of the log-likelihood ratios (associated with the emission model $E$). This constant characterizes the intrinsic difficulty of the inference task. In particular, the emission matrix $E$ directly influences $\Lambda_n$: when the likelihoods under different states (i.e., the columns of $E$) are more distinguishable, $\xi$ becomes larger, allowing for a larger choice of $\lambda$ in (34). When the emission model is known, $\xi$ can be computed analytically from (138). When the emission model is unknown or too complex, $\xi$ can instead be estimated empirically from data using (138). This parameter is related to the notion of the *error exponent* in the social learning literature; see, e.g., Khammassi et al. (2025). Now we provide an example of $\xi$ computation corresponding to the model in Section 6.1:.....''
>
> Due to the space and time limit, the detailed steps will not be shown here. However, note that due to the symmetry and the two-state structure in this example, the computation is simplified and remains easy to follow.
>
> **Linear recurrent networks in (near-)deterministic environments**
>
> Please see the section "Restrictive determinism assumption and limited evaluations" of our response to Reviewer neRB.
>
> **Second paragraph of section 5**
>
> We agree that discussing $\boldsymbol{w}_k$ before introducing near-deterministic transitions and ALF may be a bit confusing, and we will remove that sentence. In addition, by ''consistent,'' we mean that the decoding error vanishes as $\delta \to 0$, which is commonly used in adaptation theory (see Section III-C of Khammassi et al. (2025)). However, to avoid confusions, we will replace it with more accessible wording. Taking both changes into account, we will modify this paragraph as ''The ability to reproduce the true states with vanishing error is important for effective policy learning. The benefits …''
>
> **No observation history in Corollary 4.5**
>
> We will add after Corollary 4.5 “Here, only the latest observation and action are required, since there are no transient states (i.e., $r=0$).”
>
> **Relaxing assumption 5.6 (a)**
>
> Assumption 5.6 (a) is commnly used in state tracking problems. We impose it just to avoid $-\infty$ values in $\boldsymbol{w}_k$, which would complicate the analysis. In practice, zero entries in $E$ can be replaced by small positive values, so this assumption does not limit practical implementations.

---

> > ### Author Rebuttal · Reviewer_2VqL · 2026-03-31
> >
> > Thank you for your clarifications, including the about the restrictiveness of the determinism assumption, and for adding the Deep ALF comparison in the experiment of Figure 3. Could you just clarify how well it performs compared to S5 in this environment?

---

> > > ### Author Response · Authors · 2026-04-03
> > >
> > > Thank you for your time in reviewing and responding!
> > >
> > > **Clarifying Deep ALF's performance relative to S5**
> > >
> > > The updated Figure 3 is available at: https://anonymous.4open.science/r/ICML2026-Linear-Memory-Fig3-2E3B/icml2026-fig3.png.
> > >
> > > We apologize that this point was not fully illustrated in our previous response due to space limitations. We will revise the last two paragraphs of Section 6.2 as follows:
> > >
> > > ''... For the **S5-based memory**, increasing the hidden size from $12$ (the memory size of **LOF**, **ALF**, and **Deep ALF**) to $256$ allows it to match the performance of **Deep ALF**, but it still falls short of **ALF** and softmax-processed **LOF**. Notably, **Deep ALF** is more parameter-efficient than the **S5-based memory**, as its encoder has a number of parameters comparable to matrix $E$, rather than a two-layer MLP. Overall, these results show that **ALF** provides an efficient and reliable linear memory, and **Deep ALF** achieves better performance than the **S5-based memory** using fewer parameters.
> > >
> > > Moreover, when environment knowledge is partially or fully available, **Deep ALF** can leverage this information to obtain improved initialization and achieve higher returns (see Figure 8). In contrast, it remains unclear how such prior knowledge can be incorporated into existing linear recurrent networks. More detailed discussions can be found in Appendix D.2.2.’'
> > >
> > > Thank you again for the thoughtful feedback. We will make sure to reflect these changes in the final paper.

---

### Decision · Program_Chairs · 2026-04-30

**Decision:**

Accept (spotlight)

**Comment:**

All reviewers recommend acceptance, and I agree. The paper provides rigorous theoretical justification for why linear recurrent memories are effective in partially observable RL. Reviewers commended the clarity of the presentation and the soundness of the analysis. The main limitations, namely, narrow experimental scope and reliance on near-deterministic assumptions, is acknowledged but reasonable given the primarily theoretical nature of the contribution. The authors' rebuttal was thorough and resolved most concerns. I encourage the authors to incorporate the promised revisions into the final version.